

**Secondary Organic Aerosol Production from Local Emissions Dominates the**
**Organic Aerosol Budget over Seoul, South Korea, during KORUS-AQ**
Benjamin A. Nault[1,2], Pedro Campuzano-Jost[1,2], Douglas A. Day[1,2], Jason C. Schroder[1,2], Bruce
Anderson[3], Andreas J. Beyersdorf[3,*], Donald R. Blake[4], William H. Brune[5], , Yonghoon Choi[3,6],
Chelsea A. Corr[3,**], Joost A. de Gouw[1,2], Jack Dibb[7], Joshua P. DiGangi[3], Glenn S. Diskin[3], Alan
Fried[8], L. Gregory Huey[9], Michelle J. Kim[10], Christoph J. Knote[11], Kara D. Lamb[2,12], Taehyoung
Lee[13], Taehyun Park[13], Sally E. Pusede[14], Eric Scheuer[7], Kenneth L. Thornhill[3,6], Jung-Hun
Woo[15], and Jose L. Jimenez[1,2]
*Affiliations*
1. Department of Chemistry, University of Colorado, Boulder, CO, USA
2. Cooperative Institute for Research in Environmental Sciences, University of Colorado, Boulder, CO,
USA
3. NASA Langley Research Center, Hampton, Virginia, USA
4. Department of Chemistry, University of California, Irvine, Irvine, CA, USA
5. Department of Meteorology and Atmospheric Science, Pennsylvania State University, University Park,
Pennsylvania, USA
6. Science Systems and Applications, Inc., Hampton, Virginia, USA
7. Earth Systems Research Center, Institute for the Study of Earth, Oceans, and Space, University of New
Hampshire, Durham, New Hampshire, USA
8. Institute of Arctic and Alpine Research, University of Colorado, Boulder, CO, USA
9. School of Earth and Atmospheric Sciences, Georgia Institute of Technology, Atlanta, Georgia, USA
10. Division of Geological and Planetary Sciences, California Institute of Technology, Pasadena, CA, USA
11. Meteorologisches Institut, Ludwig-Maximilians-Universität München, München, Germany
12. Chemical Sciences Division, Earth System Research Laboratory, National Oceanic and Atmospheric
Administration, Boulder, CO, USA
13. Department of Environmental Science, Hankuk University of Foreign Studies, Republic of Korea
14. Department of Environmental Sciences, University of Virginia, Charlottesville, VA, USA
15. Department of Advanced Technology Fusion, Konkuk University, Seoul, Republic of Korea
* Now at: Department of Chemistry and Biochemistry, California State University, San Bernardino,
California
** Now at: USDA UV-B Monitoring and Research Program, Natural Resource Ecology
Laboratory, Colorado State University, Fort Collins, CO, USA
*Correspondence to:* J. L. Jimenez (jose.jimenez@colorado.edu)



**Abstract**
Organic aerosol (OA) is an important fraction of submicron aerosols. However, it is challenging
to predict and attribute the specific organic compounds and sources that lead to observed OA
loadings, largely due to contributions from secondary production. This is especially true for
megacities surrounded by numerous regional sources that create an OA background. Here, we
utilize *in-situ* gas and aerosol observations collected on-board the NASA DC-8 during the
NASA/NIER KORUS-AQ (KORea United States-Air Quality) campaign to investigate the
sources and hydrocarbon precursors that led to the secondary OA (SOA) production observed over
Seoul. First, we investigate the contribution of transported OA to total loadings observed over
Seoul, by using observations over the West Sea coupled to FLEXPART Lagrangian simulations.
During KORUS-AQ, the average OA loading advected into Seoul was ~1 – 3 µg sm$^{-3}$. Second,
taking this background into account, the dilution-corrected SOA concentration observed over
Seoul was ~140 µg sm$^{-3}$ ppmv$^{-1}$ at 0.5 equivalent photochemical days. This value is at the high
end of what has been observed in other megacities around the world (20–70 µg sm$^{-3}$ ppmv$^{-1}$ at 0.5
equivalent days). For the average OA concentration observed over Seoul (13 µg sm$^{-3}$), it is clear
that production of SOA from locally emitted precursors is the major source in the region. The
importance of local SOA production was supported by the following observations: (1)
FLEXPART source contribution calculations indicate any hydrocarbons with a lifetime less than
1 day, which are shown to dominate the observed SOA production, mainly originate from South
Korea. (2) SOA correlated strongly with other secondary photochemical species, including short-
lived species (formaldehyde, peroxy acetyl nitrate, sum of acyl peroxy nitrates, dihydroxy toluene,
and nitrate aerosol). (3) Results from an airborne oxidation flow reactor (OFR), flown for the first
time, show a factor of 4.5 increase in potential SOA concentrations over Seoul versus over the
West Sea, a region where background air masses that are advected into Seoul can be measured. (4)
Box model simulations reproduce SOA observed over Seoul within 15% on average, and suggest
that short-lived hydrocarbons (i.e., xylenes, trimethylbenzenes, semi- and intermediate volatility
compounds) were the main SOA precursors over Seoul. Toluene, alone, contributes 9% of the
modeled SOA over Seoul. Finally, along with these results, we use the metric $\Delta OA/\Delta CO_2$ to
examine the amount of OA produced per fuel consumed in a megacity, which shows less variability
across the world than $\Delta OA/\Delta CO$.



## 1. Introduction

Prior to 1950, 30% of the human population resided in urban areas (UNDESA, 2015). In 2007, the human population living in urban areas had increased to over 50% (making it the first time in human history that more people reside in urban than rural areas), and it is predicted that nearly 2/3 of the human population will be living in urban areas by 2050 (Monks et al., 2009; UNDESA, 2015; Baklanov et al., 2016). Urban areas are large sources of anthropogenic emissions to the atmosphere (from sources including transportation, industry, cooking, personal care products, and power produced from fossil fuels), and these emissions have important impacts on local, regional, and global air pollution, climate, and human and ecological health (Hallquist et al., 2009; Monks et al., 2009; Myhre et al., 2013; Baklanov et al., 2016; WHO, 2016; Cohen et al., 2017; Landrigan et al., 2018; McDonald et al., 2018). Effects from urban emissions are strongly modulated by the chemical evolution of the primary emissions (e.g., nitrogen oxides, hydrocarbons, and primary organic aerosols) to secondary pollutants, including secondary organic aerosols (SOA, produced from atmospheric reactions) and other aerosol (Monks et al., 2009). These emissions and their chemical by-products significantly influence hemispheric climate and air quality. They increase mortality in polluted urban areas, leading to over 3 million premature deaths annually (Lelieveld et al., 2015; Baklanov et al., 2016; WHO, 2016). Finally, the emissions and production of anthropogenic aerosol may strongly regulate cloud nucleation (Peng et al., 2014), which impacts the aerosols' direct and indirect effects on climate (Myhre et al., 2013).

Production of SOA is poorly understood (Hallquist et al., 2009; Shrivastava et al., 2017; Tsimpidi et al., 2017), including in large urban environments (Volkamer et al., 2006; de Gouw et al., 2008, 2009; Hayes et al., 2015; Woody et al., 2016; Janssen et al., 2017; Ma et al., 2017). It has been shown that a large fraction (35 – 85%) of urban fine aerosol is composed of OA (Zhang





et al., 2007; Jimenez et al., 2009), and a substantial fraction of this OA is typically SOA produced
through the chemical processing of urban hydrocarbon emissions (Kleinman et al., 2007, 2008;
Dzepina et al., 2009; Kleinman et al., 2009; DeCarlo et al., 2010; Hodzic et al., 2010; Dzepina et
al., 2011; Hersey et al., 2013; Freney et al., 2014; Zhao et al., 2014; Hayes et al., 2015; Kleinman
et al., 2016; Zhao et al., 2016; Ma et al., 2017). Also, observations indicate the majority of urban
SOA production is rapid and is nearly completed within 24 equivalent photochemical hours (a
measure of OH exposure, assuming interactions in a volume with $1.5 \times 10^6$ molecules/cm$^3$ OH
throughout a 24 h period; equivalent age enables comparison of chemistry rates across different
events or studies) (DeCarlo et al., 2010; Hayes et al., 2013; Freney et al., 2014; Hu et al., 2016;
Ortega et al., 2016; Ma et al., 2017), even during winter (Schroder et al., 2018). This consistently
rapid SOA production over urban areas around the world may be due to the short lifetime (less
than one day) of urban semi- and intermediate-volatile organic compounds (S/IVOCs), that
numerous studies suggest to be a major SOA precursors, along with aromatics (Robinson et al.,
2007; Zhao et al., 2014, 2016; Hayes et al., 2015; Ma et al., 2017; McDonald et al., 2018). S/IVOCs
and low volatility organic compounds (LVOCs—compounds produced from the photooxidation
of hydrocarbons (Robinson et al., 2007; Kroll and Seinfeld, 2008; Murphy et al., 2011; Palm et
al., 2016)) are challenging to measure due to strong interaction with inlet and instrument surfaces
(e.g., Pagonis et al., 2017), limiting our knowledge of the emission rates and concentrations of
these species in the atmosphere (Zhao et al., 2014; Hunter et al., 2017). It has also been recently
shown that historical chamber SOA yields are biased low due to unaccounted-for partitioning of
S/IVOCs to walls (Matsunaga and Ziemann, 2010; Zhang et al., 2014; Krechmer et al., 2016,
2017). These missing or under-represented compounds and low-biased yields, along with uncertain
emission inventories for SOA precursors (Shrivastava et al., 2008; Woody et al., 2016; Murphy et



al., 2017), led SOA modeling efforts over urban areas using pre-2007 models to under-predict
observed SOA concentrations (de Gouw et al., 2005, 2009; Volkamer et al., 2006; Dzepina et al.,
2009; Freney et al., 2014; Woody et al., 2016). More recent modeling efforts have achieved closure
(and sometimes over-prediction) of the observed SOA, but with some controversy about the real
causes of the increased modeled SOA (Dzepina et al., 2009; Hodzic et al., 2010; Tsimpidi et al.,
2010; Hayes et al., 2015; Cappa et al., 2016; Ma et al., 2017; McDonald et al., 2018).

Another complexity in understanding SOA production over urban areas is addressing the

contributions of transport of SOA and its gas-phase precursors. Airborne observations of SOA and
SOA precursors upwind, over, and downwind of megacities around the world (Kleinman et al.,
2007, 2008, 2009, 2016, DeCarlo et al., 2008, 2010; Bahreini et al., 2009; McMeeking et al., 2012;
Craven et al., 2013; Freney et al., 2014; Schroder et al., 2018) have constrained the role of regional
transport versus megacity emissions. In general, these studies show that there is often regional
background SOA, often due to biogenic compounds and regional pollution, transported into
megacities, but that rapid SOA production is always observed and is generally dominated by the
anthropogenic emissions from the urban area being studied.

The Seoul Metropolitan Area (SMA), as considered here, is a densely populated megacity,

extending beyond Seoul proper into the large Incheon and Gyeonggi cities. SMA has ~24 million
people, or ~50% of the South Korean population, living on ~12,000 km$^2$ of land (Park et al., 2017).
SMA has large anthropogenic emissions but is also often downwind of China, presenting the
challenge of separating local emissions and production of SOA versus transport of SOA and its
precursors from regions upwind (H. S. Kim et al., 2007; Heo et al., 2009; Kim et al., 2009, 2016,
2018; H. C. Kim et al., 2017; H. Kim et al., 2017; Jeong et al., 2017; Lee et al., 2017; Seo et al.,
2017). Most of the studies in this region have used ground-based observations and/or 3D models



to characterize the amount of aerosol, and aerosol precursors, transported to SMA and South
Korea, finding 50 to 80% of the aerosol load due to international transport in the seasons with
favorable synoptic conditions (winter and spring). Though satellites are starting to be used to
investigate transport of aerosols into SMA and South Korea (Lee et al., 2013; Park et al., 2014;
Jeong et al., 2017), retrievals typically do not provide any chemical characterization or vertical
location of the aerosol (boundary layer versus free troposphere), and are typically strongly
influenced by larger aerosols (e.g., mineral dust). Airborne observations of the upwind transport
and local production of aerosol and aerosol precursors have the potential to directly assess the
impact of transport versus local emissions in this region.

In this study, we use observations collected on board the NASA DC-8 research aircraft

during the NASA/NIER (South Korean National Institute of Environmental Research) KORean
United States Air Quality (KORUS-AQ) field campaign. These data provided the opportunity to
investigate SOA production; as well as, the role of OA and SOA precursor transport on the OA
concentration and SOA production over Seoul during the campaign. We evaluate the observed
SOA production over the SMA with source analysis models, correlation of secondary gas-phase
species with SOA, an oxidation flow reactor, and box modeling to constrain local versus transport
contributions. These results are discussed and placed into context of improving our knowledge
about SOA production and sources in urban environments.
**2. Methods**

Here, we introduce the KORUS-AQ campaign (Sect. 2.1), the key instrument for this study

(2.2), additional measurements used in the analyses (2.3), and the airborne oxidation flow reactor
(2.4). All linear fits, unless otherwise noted, use the least orthogonal distance regression fitting
method (ODR).



## 2.1 KORUS-AQ brief overview


KORUS-AQ was conducted over South Korea and the West Sea during May – June, 2016.
This study focuses on the NASA DC-8 (Aknan and Chen, 2018) observations; however, there were
numerous other measurement platforms in operation (Al-Saadi et al., 2015). The DC-8 was
stationed in the Songtan area of Pyeongtaek, South Korea, approximately 60 km south of Seoul.
The DC-8 flew 20 research flights (RF) (Figure 1; Table S1). For each RF, the DC-8 would take-
off from near Seoul, typically at 8:00 local time (LT), which is Korean Standard Time, and perform
a missed approach over Seoul Air Base, which is less than 15 km from the Seoul city center. This
pattern was typically conducted 2 more times during each flight, around 12:00 LT and prior to
landing (~15:00 LT), leading to 55 missed approaches over Seoul during the campaign. Each
missed approach involved flying near Seoul below 1000 m (above ground) for 15 – 45 minutes,
providing a large number of observations of the Seoul boundary layer. The observations collected
during this pattern, along with any other flights conducted within the coordinates defined as Seoul
(Table S2, Fig. S1) are referred to as "Seoul" below.
Briefly, the SMA is bordered by the West Sea (i.e., the Yellow Sea) and Gyeonggi Bay to
the west and forests and mountainous regions to the north, south, and east (Park et al., 2017).
Within this region, nearly 30% of the land is used for human activities, ~21% is used for cropland,
pasture, and grassland, and ~36% is forested (Park et al., 2017). During the time period of KORUS-
AQ, the wind is typically from the west to northwest, meaning that observations over the West Sea
represent typical background (inflow) air mass observations for Seoul (H. C. Kim et al., 2017).
May and June are typically characterized by low precipitation and rising temperatures prior to the
summer monsoon (Hong and Hong, 2016; H. C. Kim et al., 2017).



Besides the Seoul missed approaches, the DC-8 would fly either over the West Sea, the

Jeju jetway, or the Busan jetway at four different altitudes (nominally ~300 m, ~1000 m, ~1500

183        m, and ~7500 m above ground, depending on the height of the boundary layer, clouds, and

chemical forecasts). Many of the 3 lower elevation sampling legs also encountered significant

pollution in all regions. The approximate coordinates for these regions are also included in Table

S2. For this study, the observations over the West Sea have been split into 2 categories: "clean,"

referring to the typical conditions observed during KORUS-AQ, and "transport/polluted,"

referring to one RF (RF12, 24/May/2016) that had direct transport of pollution from the Shanghai

region over the West Sea. Also, RF11 (21/May/2016) is not included in the West Sea category, as

this flight was sampling outflow from the SMA (prevailing winds from the east).

**2.2 CU-AMS sampling and analysis**

A highly customized, high-resolution time-of-flight aerosol mass spectrometer (HR-ToF-

AMS, Aerodyne Research, Inc.) was flown on the NASA DC-8 during KORUS-AQ. Hereafter,

this instrument will be referred to as CU-AMS since there were two high-resolution AMSs on the

DC-8 during KORUS-AQ (see Sect. 2.4). Both AMSs measured non-refractory composition of

$PM_1$ (particulate matter with aerodynamic diameters less than 1 μm) (DeCarlo et al., 2006;

Canagaratna et al., 2007). The key differences between the two AMSs was the type of vaporizer

(standard for CU-AMS versus capture for the other AMS), which has been described in prior

publications (Jayne and Worsnop, 2015; Hu et al., 2017a, 2017b, 2018a, 2018b; Xu et al., 2017)

and below and in Sect. 2.4. The basic concept, and operation, of the CU-AMS for aircraft sampling

has been described elsewhere (DeCarlo et al., 2006, 2008, 2010; Dunlea et al., 2009; Cubison et

al., 2011; Kimmel et al., 2011; Schroder et al., 2018), and a brief description of other important



details follow. For detailed information on field AMS calibrations, positive matrix factorization,
photochemical clock calculations, and model setups, see Supplemental Sect. S2 – S**7**.

Ambient particles were drawn into the airplane through an NCAR High-Performance

Instrumented Airborne Platform for Environmental Research Modular Inlet (HIMIL; Stith et al.,
2009) at a constant standard (T = 273 K and P = 1013 hPa) flow rate of 9 L min$^{-1}$. The ram heating
of the inlet dried the aerosol prior to entering the airplane, and the temperature of the cabin
(typically ~10°C higher than ambient) maintained the sampling line RH to less than 40%, ensuring
the aerosol remained dry prior to entering the AMS. The sample was introduced into the AMS
aerodynamic focusing lens (Zhang et al., 2002, 2004), operated at 1.5 Torr,  through a pressure-
controlled inlet, which was operated at 433 hPa (325 Torr) (Bahreini et al., 2008). The focused
particles were then introduced after several differential pumping stages into a detection chamber,
where they impacted on an inverted cone porous tungsten vaporizer ("standard" vaporizer) held at
600°C. The non-refractory species were flash-vaporized and the vapors were ionized by 70 eV
electron ionization. Finally, the ions were extracted, and analyzed by a high-resolution time-of-
flight mass spectrometer (HTOF, Tofwerk AG). The residence time from outside the inlet to the
vaporizer was ~0.4 s in the boundary layer and ~1.0 s at 7500 m during KORUS-AQ. Unless
otherwise noted, all aerosol data reported here is at standard temperature (273 K) and pressure
(1013 hPa) (STP), leading to the notation μg sm$^{-3}$ (sm$^{-3}$ is the standard volume, in m$^3$, at STP).
Notation in scm$^{-3}$ is also at STP.

For KORUS-AQ, the CU-AMS was operated in the Fast Mass Spectrum (FMS) mode

(Kimmel et al., 2011), in order to obtain high-time resolution measurements (1 Hz) continuously.
Each FMS 1 s "run" is either collected as chopper closed (background with particle beam blocked)
or chopper open (background plus ambient particles and air) position. For KORUS-AQ, the CU-





AMS sampled with chopper closed for 6 s and chopper open for 46 s. For the remaining 8 s of the
1 min cycle, it sampled with the efficient particle time-of-flight (ePToF) mode which provides
particle sizing but with reduced sensitivity (Fig. S2). Also, once every 20 – 30 min, rapid sampling
(20 s) of outside air through a particle filter was used to ensure quality control of the instrument
background and detection limits. The average of the two background signal periods (chopper
closed) before and after the open signal was subtracted from each 1 s open measurement. In
addition to the 1 s data, we reported a 1 min data product, in which we averaged raw mass spectra
prior to fitting the high-resolution ions, leading to improved signal-to-noise (SNR) from reduced
nonlinear fitting noise (beyond the expected increased SNR from averaging in an ion-counting
noise regime). For this study, the 1 min data product is used since the additional spatial resolution
provided by the 1 s product was not required for the analysis of regional plumes. The software
packages Squirrel V1.60 and PIKA V1.20 within Igor Pro 7 (Wavemetrics) (DeCarlo et al., 2006;
Sueper, 2018) were used to analyze all AMS data.
The CU-AMS always used the "V-mode" ion path (DeCarlo et al., 2006), with a spectral
resolution ($m/\Delta m$) of 2500 at $m/z$ 44 and 2800 at $m/z$ 184. The collection efficiency (CE) for the
CU-AMS was estimated per Middlebrook et al. (2012). Calibrations of the CU-AMS are discussed
in the supplement (Sect. S2), and detections limits for the 1 min data were 26, 12, 4, 10, and 115
ng sm$^{-3}$ for $SO_4$ (sulfate), $NO_3$ (nitrate), $NH_4$ (ammonium), Chl (chloride), and OA, respectively
(Note that the charge symbol is not included for the nominally inorganic species, as organic
compounds may make (typically small) contributions to these species (e.g., organonitrates,
organosulfates, and reduced organic nitrogen species) (Huffman et al., 2009; Farmer et al., 2010).
$pNO_3$ will be used throughout the rest of the paper to represent aerosol $NO_3$ and to ensure it is not
confused with radical $NO_3$. These detection limits are estimated for every data point per Drewnick



249 et al. (2009) and remained nearly constant during each flight and throughout the campaign. It was

250 scaled by ∼×0.8 based on comparison with periodic filter blanks (Campuzano-Jost et al., 2016). It

251 was found that the scaling with the period blanks from the filters were not impacted by the length

252 of sampling outside air through the filter. The low limit of detection for these species remained

253 nearly constant during the flight by using a cryogenic pump that lowered the temperature of a

254 surface surrounding the vaporizer region to 90 K. This freezes out most background gases, and

255 provided consistently low detection limits during the flight, when other AMSs may suffer from

256 much increased detection limits for several hours into a flight due to pumping out of large initial

257 backgrounds. The 2σ accuracy for the CU-AMS of inorganic and organic species is estimated to

258 be 35% and 38%, respectively (Bahreini et al., 2009). The O/C and H/C ratios were determined

259 using the improved-ambient method (Canagaratna et al., 2015). The CU-AMS was fully

260 operational during KORUS-AQ, except for 2 hours in RF01, leading to nearly 99% data collection

261 coverage. Additional information on AMS data interpretation can be found in Jimenez et al. (2018)

262 as well as the datafile headers for the KORUS-AQ AMS data (Aknan and Chen, 2018).

263 **2.3 Oxidation flow reactor sampling and analysis**

264  The Potential Aerosol Mass (PAM) oxidation flow reactor (OFR) allows the measurement

265 of the aerosol mass that can be formed from the precursors that are present in ambient or laboratory

266 air (Kang et al., 2007; Lambe et al., 2011). The OFR has been successfully deployed in multiple

267 urban and forested locations to quantify potential SOA (Ortega et al., 2016; Palm et al., 2016,

268 2017, 2018; Kang et al., 2018). The chemical regimes and comparability to ambient results of the

269 OFR have been characterized extensively by modeling, which indicate that SOA formation

270 proceeds by chemistry similar to the atmosphere, dominated by OH oxidation under low-NO

271 conditions (Li et al., 2015; Peng et al., 2015, 2016). The sampling schematic of the OFR during



KORUS-AQ is shown in Fig. S12. Briefly, it is a 13 L (45.7 cm length OD × 19.7 cm ID)
cylindrical aluminum tubular vessel that uses two low-pressure mercury 185 and 254 nm lamps
(BHK, Inc., model no. 82-904-03) to produce OH radical through the photolysis of ambient $H_2O$,
$O_2$, and $O_3$ (R1 – R5). This mode of operation is referred to as "OFR185."

$H_2O + h\nu$ (185 nm) $\rightarrow$ OH + H                    (R1)

$O_2 + h\nu$ (185 nm) $\rightarrow$ 2 O($^3$P)                    (R2)

$O_2 + $O($^3$P)$ \rightarrow O_3$                    (R3)

$O_3 + h\nu$ (254 nm) $\rightarrow O_2 + $O($^1$D)                    (R4)

O($^1$D)$ + H_2O \rightarrow$ 2 OH                    (R5)

The KORUS-AQ study represents the first airborne operation of an OFR to our knowledge.

Unlike prior ground-based field studies (Ortega et al., 2016; Palm et al., 2016, 2017, 2018), the
UV lamps were typically maintained at one constant light setting since the OFR was sampling
more rapidly varying air masses. Since external OH reactivity (OHR) and water vapor
concentrations changed with air mass, a range of OH exposures ($OH_{exp}$) were reached inside the
OFR despite the constant photolytic flux (Peng et al., 2015). The OFR $OH_{exp}$ was calibrated using
two different methods: (1) Using the removal of ambient CO in the OFR during flight (on-line
calibration in Fig. S14a). (2) While on the ground, injecting known amounts of humidified
(multichannel Nafion drier) CO from a zero air cylinder spiked with ~2 ppmv of CO (Scott
Marrin), and varying the light intensity to produce different amounts of OH (off-line calibration in
Fig. S14b) and thus CO reactive removal. Both the off- and on-line approach yielded a calibration
factor of ×0.4 for the $OH_{exp}$, calculated using the parameterization of Peng et al. (2015), similar to
Palm et al. (2016). The $OH_{exp}$ calculated with the calibrated equation is used for periods in which
the Picarro was not sampling the OFR output. A histogram of the other key parameters used to
calculate $OH_{exp}$ with the Peng et al. (2015) equation—$H_2O(g)$ (measured by DLH) and ambient



OH reactivity (measured by ATHOS)—are shown in Fig. S15. The OFR operating conditions were
in the "Safe" zone (Peng et al., 2015, 2016), meaning that they were consistent with tropospheric
chemistry.

A key difference between the operation of this OFR during KORUS-AQ and previous field

studies (which did not use any inlet (Ortega et al., 2016; Palm et al., 2016, 2017, 2018)) was that
the gas and aerosol passed through ~1.8 m of ~4.6 mm ID stainless steel tubing at 5 vlpm
(residence time ~1.4 s through tubing). The residence time in the OFR was ~150 s.  The gas and
aerosol sample entered the OFR through a ½" press fitted stainless steel inlet that was coated in
SilcoNert (SilcoTek Co, Bellefonte, PA) and had 18 evenly spaced holes (Fig. S13), to promote
more even injection of the sample into the OFR flow cross-section (Ortega et al., 2016; Palm et
al., 2017; Mitroo et al., 2018). The gas-phase output of the OFR was sampled by an 8.25 cm
diameter Teflon ring inside the OFR connected to 1/8" Teflon tubing, and sampled by two gas
analyzers for $O_3$ (Model 205, 2B Technologies, Boulder, CO, USA) and CO (Picarro, see above).
The aerosol was sampled by a 2 mm ID stainless steel tube. A constant flow through the OFR at
all times was maintained with a bypass flow (when the CU-AMS or CO instrument were not
sampling from the OFR) to always maintain a constant residence time in the OFR. The CU-AMS
sampled from the OFR for 12 – 15 seconds every three minutes (Fig. S2). This sampling scheme
was chosen to ensure the CU-AMS had a high sampling frequency for ambient aerosol, while also
sampling the OFR once each time the air inside it was replaced (given its residence time of ~150
s).

In prior ground-based studies the OFR was placed outside, leading to the ambient and OFR

temperature being within 1 – 2°C (Ortega et al., 2016; Palm et al., 2016, 2017, 2018). During
KORUS-AQ the OFR was housed inside the DC-8 cabin, which was typically ~10°C (range 0 –





20°C) warmer than ambient air (Fig. S16). Since $(NH_4)_2SO_4$ is nonvolatile, there was little impact
on the amount of $SO_4$ entering and exiting the OFR (as confirmed when the UV lights were off
and $OH_{exp}$ was zero) (Fig. S17). For OA, which is typically a mixture of semivolatile and
nonvolatile compounds, and for $pNO_3$, which can be quite semivolatile (Huffman et al., 2009;
Cappa and Jimenez, 2010), the mass concentration exiting the OFR was significantly lower than
when entering, with lights off. This is due to evaporation of OA and $pNO_3$ at the warmer cabin
temperatures and longer residence times (~150 s). Thus, the average ratio of OA transmitted
through the OFR versus bypassing the OFR with lights off is used (slope in Fig. S17) as an
approximate correction for the amount of OA that should have exited the OFR without chemistry
when the OFR was in oxidation mode. These values were highly correlated ($R^2 = 0.94$) and did
not vary for the entire campaign, leading to confidence in the correction. This corrects for the semi-
volatile nature of ambient OA, but it does not correct for any temperature dependence of SOA
formation.

The average aerosol condensation sink (CS) inside the OFR is needed for the LVOC fate

model described in Palm et al. (2016) to compute the amount of condensable vapors that do not
form SOA in the OFR (due to residence-time limitations and surface losses), but that would be
expected to form SOA in the atmosphere. Since there were no particle sizers available to estimate
the changes in the aerosol surface area after the OFR (Palm et al., 2016), we used Eq. (1) to estimate
the average aerosol surface area in the OFR and estimated condensational sink ($CS_{est}$).
$$CS_{est}=CS_{amb}\times(\frac{\text{AMS Tot Mass Out}+\text{AMS Tot Mass In}}{2\times\text{AMS Tot Mass In}})^{2/3}\times2 \qquad (1)$$
The $CS_{amb}$ (ambient condensational sink) is calculated using the LAS and SMPS measurements.
The second term is a scaling factor to account for the observed increase in mass in OFR (with a
power of 2/3 for approximate conversion to relative surface area). The third term is a scaling factor





for relative increase in surface area due to strong nanoparticle formation in the OFR, as observed
in Los Angeles during CalNex (Ortega et al., 2016). A hygroscopic growth factor is not included
in the $CS_{est}$ (Palm et al., 2016) as the aerosol in the OFR was dry (Sect. 2.1).

One simple way to confirm the validity of the OH exposures derived from the in-field

calibrations and measurements, is to compare the observed versus modeled sulfate enhancements
in the OFR while traversing $SO_2$ plumes (Palm et al., 2016) (Fig. S18 and S19). Albeit the point-
to-point comparison is noisy as expected, we find good agreement on average between the modeled
and measured $SO_4$ enhancement (slope = 0.94), validating the quantification of the OFR for this
study. We obtained results from the OFR for all flights except RF12 (24/May/2016), when a valve
malfunction prevented measurements of $O_3$ and thus the ability to quantify OFR $OH_{exp}$. During
this flight, the Picarro was also not sampling from the OFR.
**2.4 Co-located supporting measurements used in this study**

In addition, the CU-AMS measurements, this study utilizes several co-located gas- and

aerosol-phase measurements collected on-board the DC-8.
**2.4.1 Gas-phase measurements**

NO, $NO_2$, $NO_y$, and $O_3$ were measured by the NCAR chemiluminescence instrument

(Weinheimer et al., 1994). For CO, the ambient measurements were made with the NASA Langley
tunable diode laser absorption spectroscopy (DACOM) (Sachse et al., 1987) while the
measurements for CO exiting the oxidation flow reactor (Sect. 2.3) were made with a Picarro
G2401-m. The Picarro was calibrated in flight with a WMO traceable gas standard. Gas-phase
$H_2O$ was measured with the NASA Langley open-path tunable diode laser hygrometer (DLH)
(Diskin et al., 2002). The Pennsylvania State University Airborne Tropospheric Hydrogen Oxides
Sensor (ATHOS), based on laser-induced fluorescence, measured OH, $HO_2$, and OH reactivity



(Faloona et al., 2004; Mao et al., 2009). Hydrocarbons were measured by the University of
California-Irvine whole air sampler (WAS), followed by analysis with a gas-chromatography
followed by either a flame ionization detector or mass spectrometer (Blake et al., 2003), and also
by the University of Oslo proton transfer reaction time-of-flight mass spectrometer (PTR-MS)
(Wisthaler et al., 2002; Müller et al., 2014). $SO_2$ and speciated acyl peroxy nitrates (e.g., PAN and
PPN) were measured by the Georgia Institute of Technology chemical ionization mass
spectrometer (GT-CIMS) (Huey et al., 2004; Slusher et al., 2004; S. Kim et al., 2007) The sum of
the total peroxy nitrates ($\Sigma ROONO_2$) and total alkyl and multifunctional nitrates ($\Sigma RONO_2$) were
measured by the University of California-Berkeley thermal-dissociation laser-induced
fluorescence (TD-LIF) technique (Day et al., 2002; Wooldridge et al., 2010). Formaldehyde was
measured with the University of Colorado-Boulder difference frequency absorption spectrometer
(CAMS, or Compact Atmospheric Multi-species Spectrometer) (Weibring et al., 2010; Richter et
al., 2015). Finally, HCN, $HNO_3$, and dihydroxy-toluene were measured by the California Institute
of Technology chemical ionization mass spectrometer (CIT-CIMS) (Crounse et al., 2006;
Schwantes et al., 2017).
**2.4.2 Supporting aerosol measurements**

Refractory Black carbon (BC) mass concentrations in the accumulation mode size range

measured by the NOAA Single Particle Soot Photometer (SP2) (Schwarz et al., 2013). $SO_4^{2-}$ was
measured both by the University of New Hampshire mist-chamber ion-chromatograph (MC/IC,
fine mode only) (Talbot et al., 1997) and total particulate filters, analyzed off-line with ion
chromatography (fine and coarse mode with an estimated size cut of 4 μm) (Dibb et al., 2003;
McNaughton et al., 2007; Heim et al., 2018). Besides the CU-AMS, the Hankuk University of
Foreign Studies operated an AMS onboard as well (hereinafter referred to as K-AMS); however,



using a "capture vaporizer" (see Sect. 2.2 above). Briefly, the geometry and material of the
vaporizer has been modified to reduce the amount of particle bounce, and this leads to a CE of ~1
for all ambient species, albeit with more thermal decomposition (Jayne and Worsnop, 2015; Hu et
al., 2017a, 2017b, 2018a, 2018b; Xu et al., 2017).

Finally, the physical concentration and properties of the aerosol were measured by the

NASA Langley Aerosol Research Group (LARGE). These included: (1) Size-resolved particle
number concentrations (values used to estimate surface area and volume) were measured by a TSI
Laser Aerosol Spectrometer (LAS, model 3340; TSI Inc., St. Paul, MN; calibrated with a range of
NIST traceable polystyrene latex spheres (PSL), size range 100 – 5000 nm PSL mobility diameter),
a scanning mobility particle sizer (SMPS, composed of a differential mobility analyzer, TSI model
3081 long column with custom flow system and SMPS operated using TSI software) and a CPC
(TSI model 3010, size range 10 – 200 nm PSL mobility diameter). (2) Scattering coefficients at
450, 550 and 700 nm were measured with an integrating nephelometer (TSI, Inc. model 3563) and
corrected for truncation errors per Anderson and Ogren (1998). (3) Absorption coefficients at 470,
532 and 660 nm were measured with a Particle Soot Absorption Photometer (PSAP, Radiance
Research) and corrected for filter scattering per Virkkula (2010). In order to calculate extinction,
which is used in this study, the measured Angstrom exponent was used to adjust the scattering at
550 nm to 532 nm (Ziemba et al., 2013).
**3. PM$_1$ comparisons, composition, and transport during KORUS-AQ**
**3.1 Intercomparisons of airborne PM$_1$ during KORUS-AQ**

The intercomparison of aerosol measurements less than 1 µm are summarized here and

shown in Table 1, and the full detailed intercomparison is found in SI 8. The AMS and the other



aerosol measurements agree within their combined uncertainties, similar to prior studies (DeCarlo
et al., 2008; Dunlea et al., 2009; Hayes et al., 2013; Liu et al., 2017; Schroder et al., 2018).
**Table 1.** Overview of intercomparisons for KORUS-AQ CU-AMS versus other $PM_1$
measurements. Uncertainties listed are $1\sigma$.

| Instrument Comparison | What is being Compared | Slope | $R^2$ | Combined Uncertainty of Instruments | Uncertainty of Regression Slope |
|---|---|---|---|---|---|
| MC//IC | $SO_4$ Mass | 0.95 | 0.76 | ±20% | ±1% |
| Filters | $SO_4$ Mass | 0.80 | 0.86 | ±24% | ±2% |
| AMS Scatter Plot (Total Campaign) | Total $PM_1$ Mass/CE | 0.95 | | ±27% | ±1% |
| AMS Scatter Plot (Lower $PM_1$ Sizes)[a] | Mass/CE/ Transmission | 1.02 | 0.91 | ±27% | ±1% |
| AMS Scatter Plot (Higher $PM_1$ Sizes)[b] | Mass/ Transmission | 0.84 | 0.82 | ±27% | ±1% |
| Extinction | Total $PM_1$ Mass to 532 nm Extinction | 6.00 | 0.87 | ±31% | ±3% |
| LAS (all data) | $PM_1$ Volume | 1.56 | 0.86 | ±43% | ±1% |
| LAS (Conc. Filter)[c] | $PM_1$ Volume | 1.19 | 0.91 | ±43% | ±1% |
| LAS (Mass Filter)[c] | $PM_1$ Volume | 1.00 | 0.79 | ±43% | ±1% |

[a]Comparison of K-AMS and CU-AMS for RFs 1 – 9, 11, 15, 19. [b]Comparison of K-AMS and CU-
AMS for RFs 10, 12 – 14, 16 – 18, 20. [c]Using the lower concentration and mass filter. See SI 8
and Fig. S27.
There are two comparisons that show lower agreement. The slope between AMS $SO_4$

versus filter $SO_4$ is 20% lower than unity. As previously mentioned, the filters collect aerosols
with diameters up to 4 μm, and Heim et al. (2018) concluded that the difference was due to
supermicron $SO_4^{2-}$ throughout the campaign from the transport of dust from continental Asia to
South Korea. Thus, the differences in the diameter cut-off between the AMS and filters and
observations of supermicron $SO_4^{2-}$ are the likely cause of the lower slope. The comparison of
calculated volume from AMS plus BC versus calculated volume from LAS indicate that the AMS
and BC calculated volume is higher. Part of the reason could be that LAS vaporizes BC (Kupc et
al., 2018); however, as shown in Figure 1, BC accounts for a small fraction of $PM_1$. As discussed
in SI 8, the LAS detector saturated at high particle number concentrations/high mass
concentrations. This has also been observed by Liu et al. (2017) in a prior airborne campaign with



the same instruments. Comparing the LAS and AMS plus BC volume at lower total mass
concentrations/particle number concentrations, the difference is reduced to ~10%, which is well
within the combined uncertainties.
**3.2 PM$_1$ concentration and composition over South Korea during KORUS-AQ**
We briefly describe the PM$_1$ composition observed over South Korea during the campaign
and compare it to prior observations in the same region and for other large urban areas around the
world. First, the comparison of PM$_1$ in the boundary layer (estimated to be ~600 m from
temperature profiles measured on the DC-8 during the entire campaign, as shown in Figure 1) in
the KORUS-AQ domain is discussed. As shown in Figure 1, the highest average PM$_1$ during
KORUS-AQ were observed over the West Sea during the "transport/polluted" research flight
(RF12, 24/May/2016), at 39 $\mu$g sm$^{-3}$, and over Seoul, at 31 $\mu$g sm$^{-3}$ (average of all flights over
Seoul). However, during the latter half of the mission, PM$_1$ was regularly greater than 60 $\mu$g sm$^{-3}$,
and as high as 100 $\mu$g sm$^{-3}$, over Seoul. During the rest of the flights, the average mass
concentration over the West Sea was a factor of 3 lower (13 $\mu$g sm$^{-3}$). Also, the Seoul and West
Sea PM$_1$ composition was different, where SO$_4$ and more-oxidized oxidized organic aerosol (MO-
OOA) dominated the West Sea PM$_1$ budget, indicative of transported, aged chemistry (Dunlea et
al., 2009; Lee et al., 2015). Also, the O/C ratio for the West Sea was 0.84 (WS Clean) to 0.88 (WS
polluted). Seoul showed higher fractions of less-oxidized OOA (LO-OOA) than MO-OOA, and
higher pNO$_3$ than SO$_4$, which is more typical of fresher, urban chemistry, with an average O/C
ratio of 0.70 (DeCarlo et al., 2008; Hennigan et al., 2008; Hayes et al., 2013; H. Kim et al., 2017;
Kim et al., 2018). The eastern side of South Korea had lower average PM$_1$ than observed over
Seoul. This region of South Korea is not as highly populated as around Seoul, reducing the sources
and production of PM$_1$ and is more representative of PM$_1$ background pollution/transport across



the country. The average $PM_1$ observed over the Jeju jetway was similar to what was observed
over Seoul. Also, this area had similar contributions from hydrocarbon-like organic aerosol (HOA)
and $pNO_3$ as Seoul, indicating local emissions, including industry, along with
transport/background, are impacting the $PM_1$ composition (e.g., Hayes et al., 2013). This part of
South Korea has some large population centers (e.g., Gwangju and Jeonju) and power plants (e.g.,
Boryeong Power Station), which are consistent with the observed impact.

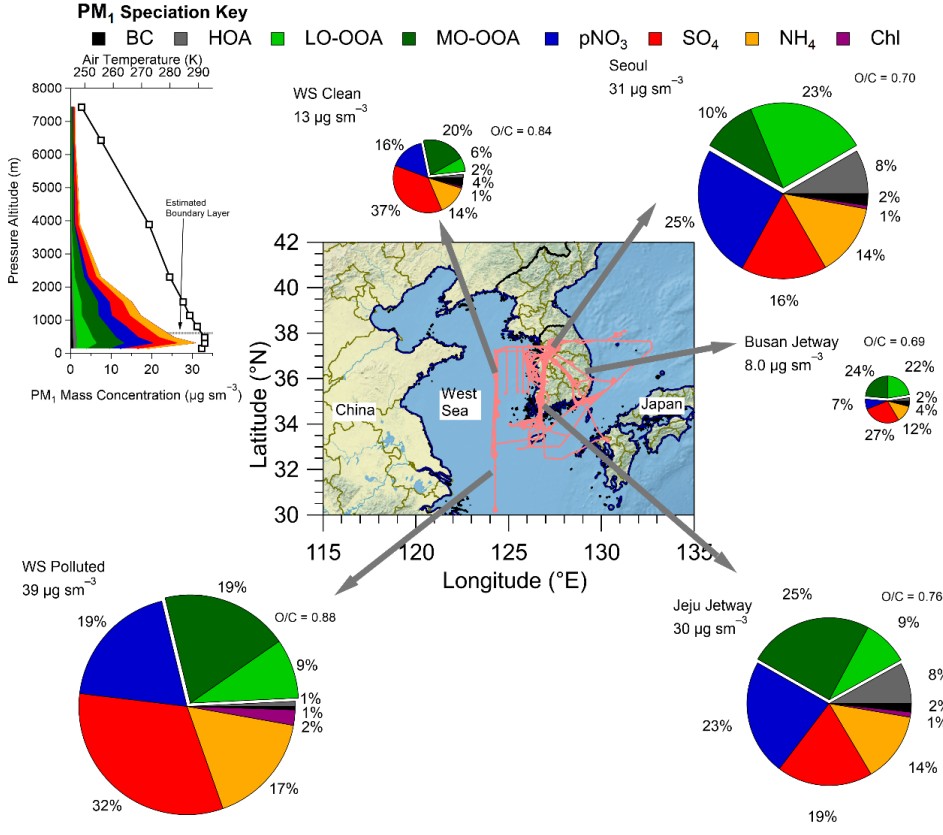


**Figure 1.** Pie charts of the average boundary layer $PM_1$ composition by the different regions
(defined in Table S2) sampled over South Korea during the campaign. The flight paths are shown
in light red. The Busan jetway had no measurable Chl; therefore, Chl is not included in the pie
chart. The pie charts area is proportional to $PM_1$ in each region. The average O/C for the OA is
shown by each OA section in the pie charts. The map shows the DC-8 flight paths throughout





KORUS-AQ. The average vertical profile of $PM_1$ species (along with temperature and continental
PBL height) over all of South Korea is shown in the upper left.

The average $PM_1$ observed over Seoul during KORUS-AQ was similar to the mass

concentration measured in Seoul in previous years (37, 38, 37, 27, and 22 µg m$^{-3}$ for Choi et al.,
2012, Kim et al., 2007, H. C. Kim et al., 2016, Park et al., 2012, and Kim and Zhang, 2017,
respectively). Also, the average $PM_1$ over the West Sea during clean conditions (13 µg sm$^{-3}$) is in
line with what has been reported over Baengnyeong Island (Lee et al., 2015), located west of South
Korea in the West Sea (37°58'00" N, 124°37'04" S). Finally, the $PM_{2.5}$ mass concentrations have
remained nearly constant for the last ~20 years (OECD, 2018).

The $PM_1$ composition over Seoul is dominated by $pNO_3$ and SOA, similar to what was

observed on the ground during the same time period (Kim et al., 2018). The composition over
Seoul is more similar to what has been observed over Mexico City during MILAGRO (DeCarlo
et al., 2008), and in Los Angeles during CalNex (Hayes et al., 2013) than observed over large
urban areas in Asia (Hu et al., 2016).

As the differences in $PM_1$ composition for the different regions mostly occurred in the

boundary layer, we show the average $PM_1$ profile observed during all of KORUS-AQ in the inset
of Figure 1, and the fractional contribution to the profile in Fig. S32. At low altitudes, the $PM_1$
mass is dominated by secondary $PM_1$ species that compose the largest fractions in the pie charts
in Figure 1 (LO-OOA, MO-OOA, $pNO_3$, and $SO_4$). The fractional contributions of $SO_4$ and MO-
OOA to $PM_1$ mass increase with altitude and become dominant above ~4 km, representative of
more aged aerosol, away from sources (e.g., Dunlea et al., 2009).



### 3.3 Analysis of background and transport influence on $PM_1$

Transport of aerosols and aerosol precursors from distant sources creates a larger-scale background that needs to be quantified in order to understand the impact of local emissions on aerosol production. Prior studies have shown the potential impact of long-distance transport in creating a background aerosol mass over Seoul (H. S. Kim et al., 2007; Heo et al., 2009; Kim et al., 2009, 2016, 2018; H. C. Kim et al., 2017; H. Kim et al., 2017; Jeong et al., 2017; Lee et al., 2017; Seo et al., 2017). To investigate the influence of background and transported air to Seoul, the FLEXPART Lagrangian model with WRF winds and meteorology is used. The application of FLEXPART to this study is described in SI 7. Briefly, the model uses back trajectories from the point where the DC-8 was sampling and calculates the amount of CO and $NO_2$ contributed by different emission regions for each sampled air parcel.

The CO concentration measured during KORUS-AQ can be described by Eq. (2). The average CO mixing ratios observed during the campaign are used.

$$CO_{ambient}=CO_{hem.\ bckg.}+CO_{foreign}+CO_{South\ Korean} \qquad (2)$$

Here, $CO_{hem.\ bckg.}$ is the hemispheric background of CO. In FLEXPART, the foreign emissions are from China, Hong Kong, Japan, Laos, Macau, Myanmar, Mongolia, North Korea, Russia, Taiwan and Vietnam. FLEXPART does not include the CO hemispheric background; therefore, that term is estimated from upwind sites (Mt. Waliguan, China and Ulann Uul, Mongolia) (Novelli et al., 2017) to be 140 ppbv CO. The West Sea is the simplest case, as FLEXPART predicted that all the CO originated from the foreign sources listed above (Figure 2a). Thus, $CO_{ambient}$ in the West Sea can be attributed to 140 ppbv $CO_{hem.\ bckg.}$ and 125 ppbv $CO_{foreign}$ (Figure 2b). The advection of CO from the West Sea to Seoul will lead to dilution and mixing of the $CO_{foreign}$ with air containing only the $CO_{hem.\ bckg.}$. With an average wind speed of 4 m/s over the West Sea, and a distance of



~300 km, the air takes ~1 actual day to move from where the DC-8 sampled over the West Sea to
Seoul. The results from FLEXPART are used to estimate the dilution rate, ~0.7 day$^{-1}$, comparable
to the values determined in prior studies (McKeen et al., 1996; Price et al., 2004; Arnold et al.,
2007; Dzepina et al., 2011; Fried et al., 2011). Thus, after 1 day of advection, ~60 ppbv $CO_{foreign}$
is expected over Seoul (Figure 2c); thus, with Eq. (2), the total CO background ($CO_{foreign}$ + $CO_{hem.}$
$_{bckg.}$) is 200 ppbv, and the remainder of the observed ambient CO is attributed to local South Korean
emissions (on average, 165 ppbv CO). Finally, results from FLEXPART show that the $CO_{foreign}$
contribution (Figure 2d) remained nearly constant throughout the campaign at all observed
photochemical ages. Therefore, 200 ppbv CO background for observations over Seoul will be used
throughout this study.

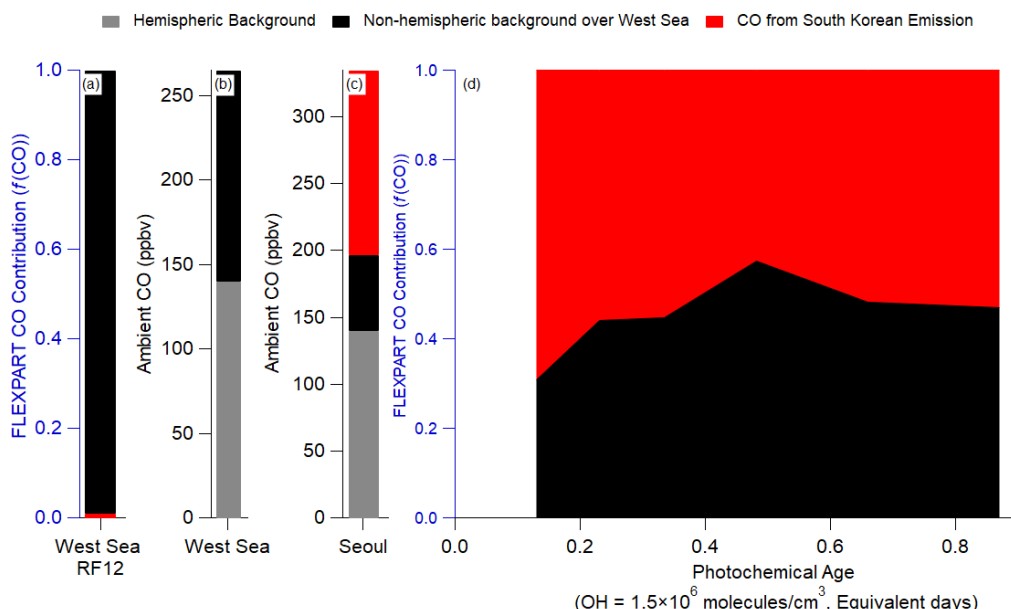

**Figure 2.** Note differences in labels and values for each y-axis. Also, note that FLEXPART does
not include hemispheric background; therefore, it is not included in the figure. (a) Fractional
contribution of foreign versus South Korean CO emission over the West Sea from FLEXPART.
(b) Estimated measured partitioning of the average CO observed over the West Sea. (c) Same as
(b), but for over Seoul. (d) Same as (a), but for over Seoul. For all panels, CO does not include any
chemical losses or production.




From the observed dilution-corrected OA concentrations (OA concentration divided by
background subtracted CO mixing ratios) over the West Sea (13 and 40 µg sm$^{-3}$ ppmv$^{-1}$ for clean
West Sea and RF12 West Sea, respectively), the CO$_{foreign}$ over Seoul would correspond to 1 – 3
µg sm$^{-3}$ OA background (Eq. (3)).
$$OA_{background}=CO_{foreign}\times(\frac{OA}{CO})_{foreign} \qquad (3)$$
The upper limit will be used for the remainder of the study. The corresponding observed
background values for HOA, LO-OOA, and MO-OOA are 0, 1±1, and 2±2 µg sm$^{-3}$, respectively.
Finally, the background for key gas-phase and aerosol species (which are discussed below) are
1.05±0.28 (CH$_2$O), 69±5 (O$_x$), 0.25±0.06 (PAN), and 0.30±0.10 (ΣROONO$_2$) ppbv, 0.44±0.34
(Dihydroxy toluene) pptv, and 2±2 µg sm$^{-3}$ (pNO$_3$). Thus, the increase in OA mass concentration
from the background values (3 µg sm$^{-3}$) to average Seoul values (13 µg sm$^{-3}$) must be due to South
Korean emissions of POA and production of SOA.

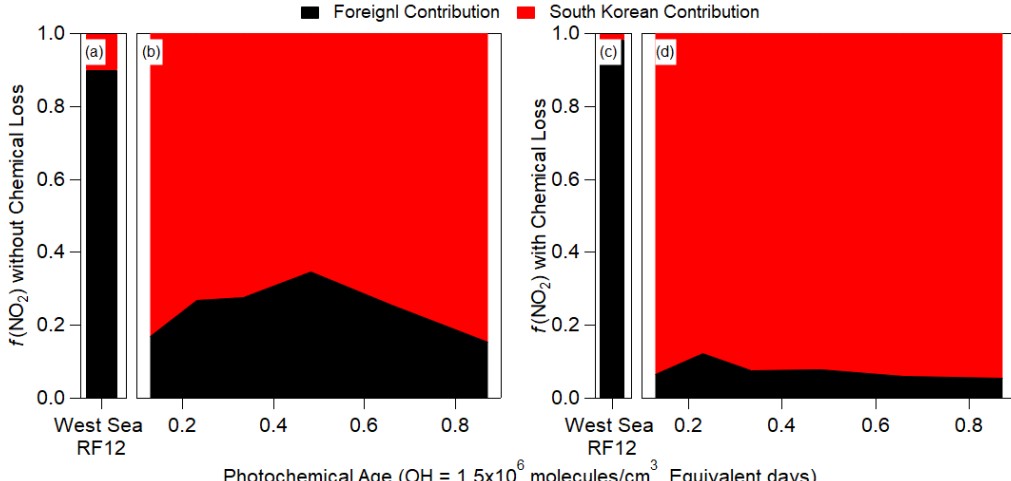


**Figure 3.** Binned fractional contribution (South Korea/(South Korea + Foreign)) of the
FLEXPART, sampled from aircraft position for contributions to (a and b) NO$_2$ (without any
chemical losses) and (c and d) NO$_2$ (with chemical losses (τ = 1 day)) versus the observed (aircraft)
photochemical age. For (a) and (c), the West Sea bars are the average fractional contributions for
RF12.



The contribution of foreign versus South Korean emissions of $NO_2$ from FLEXPART over
Seoul and West Sea (Figure 3) is analyzed next. $NO_2$ is investigated since it has a photochemical
lifetime of ~1 day (at OH ≈ $1.5 \times 10^6$ molecules/cm³). This lifetime is similar to short-lived
hydrocarbons (e.g., xylene, S/IVOC, etc.) that are thought to dominate urban SOA production in
this campaign and other megacities (de Gouw et al., 2005; Kleinman et al., 2007, 2008; DeCarlo
et al., 2010; Wood et al., 2010; Hayes et al., 2013, 2015, Hu et al., 2013, 2016; Ortega et al., 2016;
Ma et al., 2017; Schroder et al., 2018). In general, $f(NO_2)_{foreign}$ and $f(NO_2)_{local}$ (with and without
photochemical loss included in the FLEXPART model runs) is quite constant with photochemical
age (Figure 3), like CO.  Unlike CO, the contribution of local $NO_2$ is ~70% (if photochemical
removal is not included) and ~90% (if photochemical removal is included). This strongly suggests
that most short-lived hydrocarbons over Seoul, which are believed to dominate urban SOA
production, are dominated by South Korean emissions and not transport from foreign sources.
**4. SOA production over the Seoul Metropolitan Area**
**4.1 SOA production over Seoul during KORUS-AQ**
The conceptual model for analysis of photochemical SOA production over and downwind
of megacities has been discussed in detail in de Gouw (2005) and DeCarlo et al. (2010) and
subsequent studies (Hayes et al., 2013; Hu et al., 2013, 2016; Freney et al., 2014; Schroder et al.,
2018). An air mass with "background" values of OA and CO is advected over a megacity area,
where fresh emissions of POA, SOA precursors, and CO are emitted into the air mass. The SOA
precursors will oxidize to produce SOA and undergo dilution with the surrounding background air
masses. To correct for this dilution effect, the change of OA over background OA ($\Delta OA = OA -$
background OA) is divided by the change of CO over background CO ($\Delta CO = CO -$ background
CO), and this term is the dilution-corrected concentration. CO has been used in prior studies as a



surrogate for primary pollution emissions as this compound has high signal-to-noise between
urban plumes and background and a long photochemical lifetime (meaning minimal CO is lost due
to chemistry or produced from VOC oxidation over a ~1 day timescale (Griffin et al., 2007)) (de
Gouw et al., 2005; DeCarlo et al., 2010). Finally, $\Delta OA/\Delta CO$ is plotted versus estimated
photochemical age. The photochemical age approximately accounts for the chemical evolution of
precursors either into products which can be estimated from the time evolution of $NO_x/NO_y$, or the
differences in removal rates of two hydrocarbons (o-xylene or m+p-xylene to ethylbenzene). See
SI 5 for more information about the calculation of the photochemical age. The potential impact of
SOA precursors being advected into Seoul is addressed in Sect. 4.4.
Throughout the paper, the estimated photochemical age from $NO_x/NO_y$ will be used as this
measurement has higher temporal coverage, and an emissions ratio is not needed to calculate the
photochemical age (SI 5), but note that ages estimated from hydrocarbon-based clocks are
consistent. For photochemical ages greater than 1 day (measurements over the West Sea), the
aromatic photochemical clock is used (SI 5) as the $NO_x/NO_y$ clock does not work well past 1
equivalent day (SI 5). As discussed in Parrish et al. (2007), compounds used to calculate
photochemical ages should have lifetimes on the order of the range in ages to be quantified. For
example, photochemical age over Seoul is less than 1 equivalent day, which is equivalent to the
$NO_x$ lifetime; whereas, the photochemical age over the West Sea is expected to be a few equivalent
days, which is bracketed by the lifetimes of benzene and toluene.





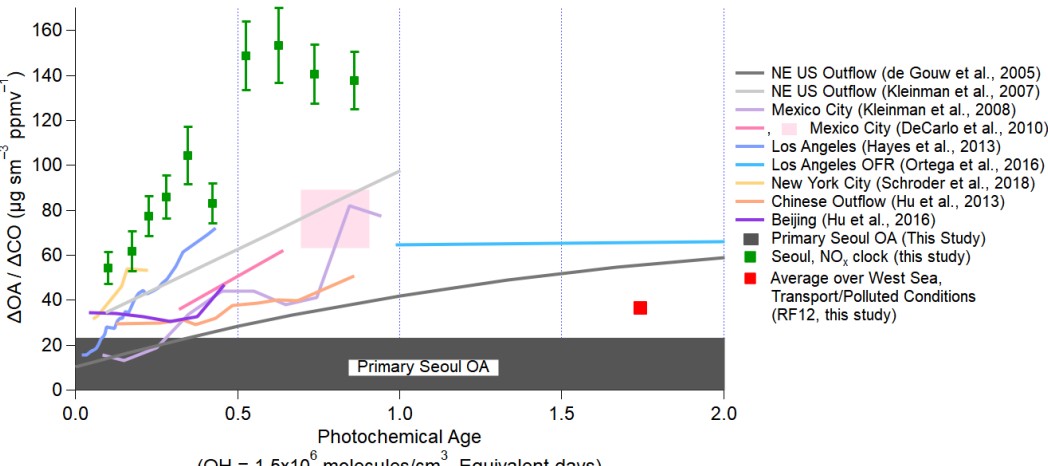


**Figure 4.** Evolution of dilution-corrected OA versus equivalent photochemical age (days), where
$\Delta OA = OA -$ background OA and $\Delta CO = CO -$ background CO, during KORUS-AQ. Over Seoul
and the West Sea, the CO background is 200 (hemispheric plus foreign) and 140 ppbv (hemispheric
only), respectively. The vertical error bars for the observations during KORUS-AQ are the
standard error of $\Delta OA/\Delta CO$ for each bin. Photochemical age is determined by the $NO_x/NO_y$. The
dark grey bar at the bottom (22 $\mu g\ sm^{-3}\ ppmv^{-1}$) is the observed POA over Seoul during the
campaign. Observations from other megacities (de Gouw et al., 2005; Kleinman et al., 2007, 2008;
DeCarlo et al., 2010; Hayes et al., 2013; Hu et al., 2013, 2016; Ortega et al., 2016; Schroder et al.,
2018) are also shown, as lines, for comparison, and have been updated, as described in Schroder
et al. (2018).

Similar to prior studies, OA over Seoul increased rapidly within the first photochemical

equivalent day from the emission source (Figure 4). The dilution-corrected SOA production is very

rapid for photochemical ages less than 0.7 equivalent days (note that this would be only 4 actual

hours of exposure for average OH concentration of $6\times10^6$ molecules/cm$^3$ observed during this

campaign). After that time, the dilution-corrected OA plateaus and remains nearly constant.

George et al. (2008) and Ortega et al. (2016) found that after ~4 – 5 equivalent days, OH

heterogeneous reactions start fragmenting the compounds in SOA, leading to a reduction in the

dilution-corrected OA mass with age. Compared to prior megacity studies, the dilution-corrected

OA produced over Seoul is between 40 – 80 $\mu g\ sm^{-3}\ ppmv^{-1}$ higher at ~0.5 equivalent days of

photochemical aging and ~70 $\mu g\ sm^{-3}\ ppmv^{-1}$ higher than in Chinese megacities (Hu et al., 2013,



2016). The strong SOA production over Seoul is similar to that observed in other relatively isolated
megacities (e.g., Los Angeles and Mexico City). It also appears not to be significantly influenced
by the outflow of upwind Chinese megacities, since Seoul SOA formation is very rapid and occurs
much faster than air mass transport from those megacities, and since the dilution-corrected
production is much larger in Seoul than in Chinese megacities.
Finally, these results do not depend on the assumed CO background. As shown in Fig. S34,
the OA mass concentration increases versus CO mixing ratios as photochemical age increase, and
in Fig. S35, even assuming a lower CO background (140 ppbv), the dilution-corrected OA
concentration is still ~100 µg sm$^{-3}$ ppmv$^{-1}$. This value is still higher than what has been observed
in prior cities (Figure 4). If the CO background is higher than assumed here (200 pbbv), the OA
production would be even higher.
To further investigate the potential influence from upwind megacities, we compare
ΔOA/ΔCO over the West Sea versus that over Seoul (Figure 4). The average ΔOA/ΔCO over the
West Sea during the "transport/polluted" conditions (RF12, 24/May/2016) is ~40 µg sm$^{-3}$ ppmv$^{-1}$.
This value is similar to the upper limit values observed in Beijing (Hu et al., 2016) and Changdao
(Chinese outflow) (Hu et al., 2013). This suggests that for Chinese outflow, the maximum dilution-
corrected OA concentration is ~40 µg sm$^{-3}$ ppmv$^{-1}$. This value has already been subtracted from
the observations over Seoul. Finding OA concentrations greater than POA concentrations at the
youngest photochemical ages may be due to (1) very rapid SOA production; (2) sunrise occurring
3 – 4 hours (sunrise between 5:10 – 5:30 LT) prior to sampling air over Seoul in the morning; and,
(3) the imperfect characterization provided by the average photochemical age when fresh
emissions have been recently injected into an air parcel.





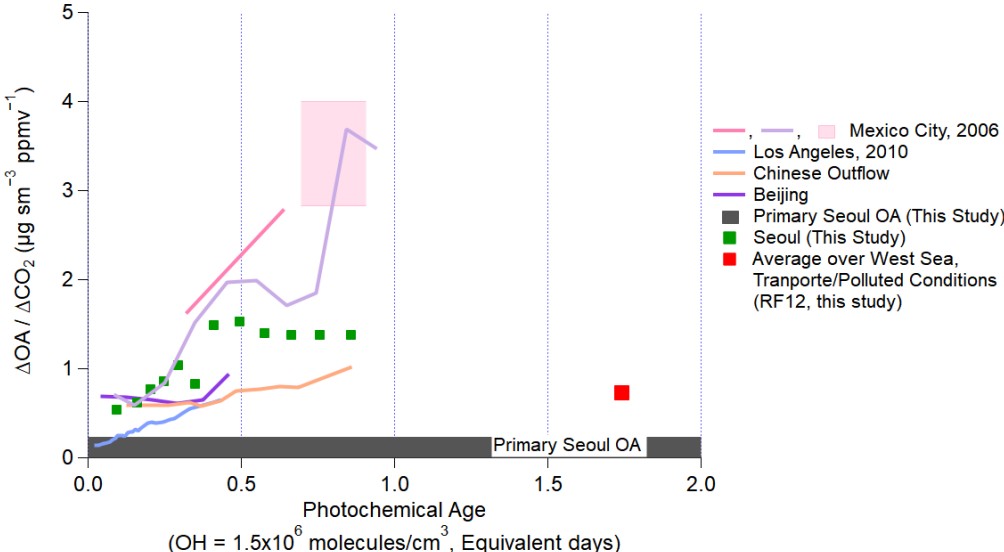

**Figure 5.** Same as Figure 4, but normalized by $\Delta CO_2$. The ratios to $CO_2$ are calculated using $\Delta CO/\Delta CO_2$ emissions ratios from prior studies in each megacity that occur during the same campaign or for the same time of year, since direct measurements of the $CO_2$ enhancements above background during the aircraft studies of each megacity are very challenging. (Table S4).

Here, we introduce another dilution-correction method to investigate SOA production over a megacity—$\Delta OA/\Delta CO_2$ (Figure 5). $\Delta OA/\Delta CO_2$ is a way to investigate the amount of OA produced per unit mass of fuel burned in each megacity. Note that although some SOA precursors are not emitted from combustion sources, such as volatile consumer products (McDonald et al., 2018), one can still define this ratio in an average sense for each megacity. It has been used previously for laboratory experiments (e.g., Gordon et al., 2013; Platt et al., 2013, 2017) and biomass burning (e.g., Akagi et al., 2012; Collier et al., 2016); however, to the best of the authors' knowledge, it has not been used for SOA production over a megacity. As noted above, CO has been typically used instead, given that it is always measured in pollution studies, and it typically has a higher signal-to-background ratio than $CO_2$ in urban areas. Also, during spring and summer, $CO_2$ is taken up by plants, which can reduce its signal-to-background ratio. However, the ratio of



other gases to CO can vary between urban areas depending on the average combustion efficiency
of the dominant sources (Silva et al., 2013). $CO_2$ better accounts for fuel consumption in an urban
area (Vay et al., 2009; Tang et al., 2018). Multiple recent studies have reported average emissions
ratios for different megacities based on high precision measurements of $\Delta CO/\Delta CO_2$ (Vay et al.,
2009; Wang et al., 2010; Peischl et al., 2013; Silva et al., 2013; Tohjima et al., 2014; Tang et al.,
2018). These results provide an ability to convert the $\Delta OA/\Delta CO$ determined in prior studies to
$\Delta OA/\Delta CO_2$ (Table S4). We find that for most of the megacities studied, $\Delta OA/\Delta CO_2$ is very
similar, though Mexico City and Seoul show higher values (approximately factor of 2). The range
of observed $\Delta OA/\Delta CO_2$ versus photochemical age is narrower, compared to the spread for all
megacities observed for $\Delta OA/\Delta CO$. Both analyses suggest that Seoul has larger relative emissions
of SOA precursors compared to other megacities, which could be targeted for air quality
improvement. However, more observations across other megacities and additional comparative
analyses would be beneficial.
**4.2 Composition-based analysis of the foreign versus South Korean contribution to SOA**
**precursors and SOA over Seoul**
**4.2.1 Evolution of oxygenated organic aerosol over Seoul**
Here, we focus on the positive matrix factorization (PMF) (Ulbrich et al., 2009) factors for
OA resolved during KORUS-AQ, whose evolution over Seoul is shown in Figure 6. Total OOA
(LO-OOA plus MO-OOA) is used as a surrogate of total SOA. The fractional contribution of these
two factors can be found in Fig. S36. Rapid production of OOA is observed, accounting for all of
the observed growth in total OA over Seoul. LO-OOA, overall, is slightly more abundant than
MO-OOA (Fig. S36). LO-OOA has lower O/C compared to MO-OOA (Fig. S10); thus, the faster
production of LO-OOA likely represents the less oxidized OOA produced from the photooxidation



of SOA precursors (Hayes et al., 2013; Freney et al., 2014; Hu et al., 2016; Kim et al., 2018), while
MO-OOA may represent the more oxidized species or those formed from later generations of
oxidation (Robinson et al., 2007; Miracolo et al., 2010; Tritscher et al., 2011; Ortega et al., 2016;
Sato et al., 2017; Schwantes et al., 2017).

The PMF factors have very different dilution-corrected concentrations over the West Sea

during the "transport/polluted" event (Figure 6a). All ΔPMF factors/ΔCO show much lower values
than for aged air over Seoul. The discontinuity between the three factors between Seoul and West
Sea indicate that transported OA, and transported SOA production, has limited impact on the OA
over Seoul.

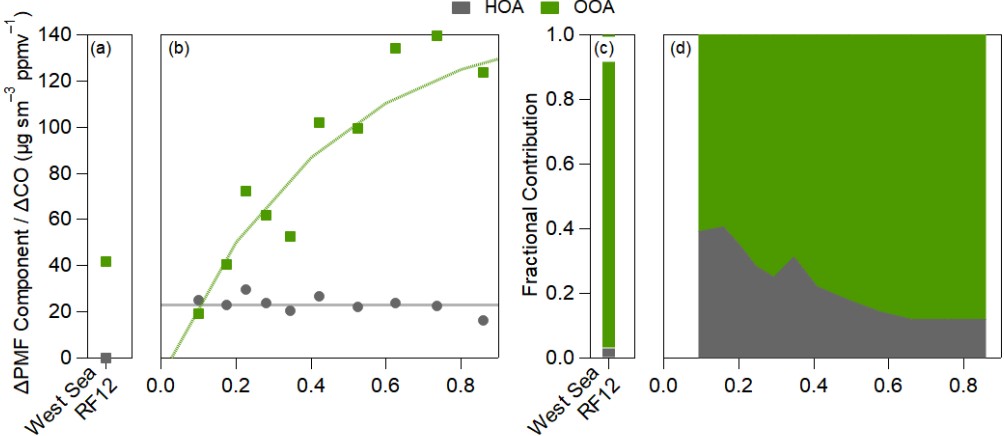


**Figure 6.** Same as Figure 4, but for the PMF results of the OA evolution over (a) West Sea during
the polluted event (West Sea RF12 at ~1.75 equivalent days) and (b) over Seoul. For (b), fit to
HOA and OOA are 23 µg sm$^{-3}$ ppmv$^{-1}$ and 150×(1−exp(−2.3×eq. day), respectively. For the OOA
equation, 150 equals the max SOA. Fractional contribution of the PMF factors over (c) West Sea
(RF12) and (d) over Seoul.

The slope of ΔHOA/ΔCO versus age was nearly zero µg sm$^{-3}$ ppmv$^{-1}$ equivalent day$^{-1}$,

indicating minimal evolution within these timescales. There are mixed results on whether HOA
changes with photochemical age (DeCarlo et al., 2010; Hayes et al., 2013; Hu et al., 2013, 2016;



Freney et al., 2014; Schroder et al., 2018); however, due to the uncertainty that comes from the
CO background, it is difficult to determine whether these changes are real or not.

At the lowest photochemical ages, HOA is ~35% of the total $\Delta OA/\Delta CO$ (Figure 6d). Since

HOA remains approximately constant with age while OOA rapidly increases (Figure 6b), the HOA
contribution to total $\Delta OA/\Delta CO$ decreases to ~10% after ~1 equivalent day. It has been observed
in prior urban campaigns that HOA contributes 10 – 50% and total POA (HOA + other primary
OA factors in AMS) contributes 30 – 60% (Aiken et al., 2009; DeCarlo et al., 2010; Hayes et al.,
2013; Crippa et al., 2014; Hu et al., 2016; H. Kim et al., 2017; Kim et al., 2018). The fractional
contribution of HOA in Seoul is within this range.

A more detailed discussion of the behavior of AMS OA source tracers can be found in SI

11. In general, the AMS OA source tracers behave similarly to other urban campaigns (e.g., Hayes
et al., 2013; Freney et al., 2014). Some dilute biomass burning OA was evident, but this source
was not major most of the KORUS-AQ. Similarly, isoprene oxidation chemistry was not a major
contributor to SOA during this campaign.
**4.2.2 Correlation of SOA versus other fast photochemical products**

Results above support that a major fraction of the SOA observed over Seoul is rapidly

produced through photooxidation of South Korean SOA precursors. To further evaluate this result,
we analyze the correlation of OOA with other secondary species known to be rapidly produced
through photooxidation of organic precursors.

The other secondary species used in this study are odd oxygen ($O_x$), formaldehyde ($CH_2O$),

peroxy acetyl nitrate (PAN), the sum of all acyl peroxy nitrates ($\Sigma ROONO_2$), and $pNO_3$. $O_x$
(approximated as $NO_2 + O_3$) is used instead of $O_3$ to account for titration of $O_3$ in the presence of
fresh NO emissions. Prior studies have used $O_x$ to provide insights into SOA production (Herndon



et al., 2008; Wood et al., 2010; Hayes et al., 2013; Morino et al., 2014; Zhang et al., 2015; Hu et
al., 2016) since $O_x$ has a similar lifetime to SOA (~1 week) (Jacob, 2000; Goldberg et al., 2015;
Hodzic et al., 2015; Ortega et al., 2016), and $O_x$ is also produced through the photooxidation of
organic compounds. However, since both $O_x$ and SOA have longer lifetimes, the correlation
observed between these two species may have a contribution from transport of polluted air masses.
To reduce the influence of transport on this analysis, the correlation of OOA with $CH_2O$, PAN,
and $\Sigma ROONO_2$ is also investigated. The benefit of these species is that they have estimated
lifetimes of less than 3 hours during daytime in KORUS-AQ (typical temperature for transported
air 17°C). Also, it has been shown that dilution-corrected $pNO_3$ decreases rapidly from urban
centers, possibly due to dilution with surrounding air low in $HNO_3$ and $NH_3$ and irreversible uptake
of $HNO_3$ onto coarser particles (e.g., DeCarlo et al., 2008).

Example time series of OOA with $O_x$, $CH_2O$, PAN, $\Sigma ROONO_2$, and $pNO_3$ during three

different afternoon Seoul overpasses are shown in Figure 7a. All gas and aerosol species exhibit
similar behavior, indicating that these species are undergoing photochemical production during
these afternoon passes, similar to what has been observed in other urban environments during the
afternoon (Perring et al., 2010; Fried et al., 2011; Parrish et al., 2012; Hayes et al., 2013; Zhang et
al., 2015). OOA also tracks the evolution of these species, consistent with OOA also being a
secondary product from hydrocarbon photooxidation.

Analyzing the entire KORUS-AQ campaign, correlations with $R^2 > 0.50$ are observed

between OOA and $O_x$, $CH_2O$, PAN, $\Sigma ROONO_2$, and $pNO_3$ for the overpass observations after
12:00 LT (Figure 7b-f). These correlations for these secondary species produced through the
oxidation of hydrocarbons, in the afternoon, when photochemical production dominates over

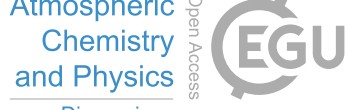



mixing and losses, further supports that the OOA production observed in Figure 4 and 6 is
dominated by the photochemistry of locally emitted hydrocarbons.

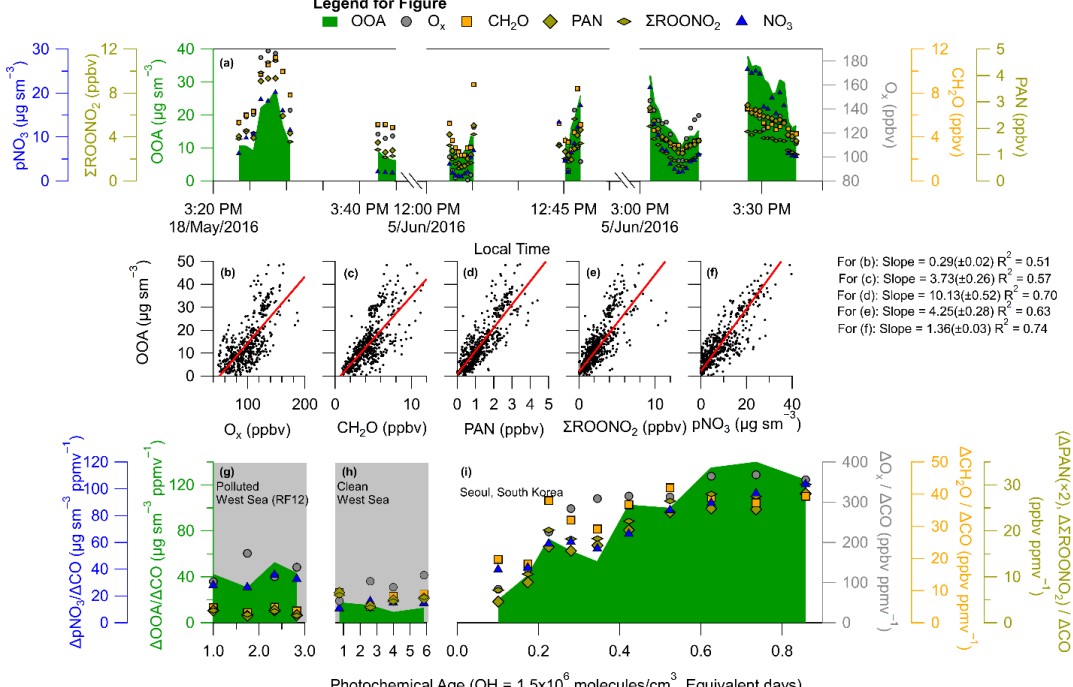


**Figure 7.** (a) Time series of OOA (OOA = LO-OOA + MO-OOA), $O_x$, $CH_2O$, PAN, $\Sigma ROONO_2$,
and $pNO_3$ during RF09 (18/May/2016), and RF18 (05/June/2016) noon and afternoon overpasses.
Gaps in time series correspond to climbing out of the boundary layer. OOA versus (b) $O_x$ ($O_x$ =
$O_3 + NO_2$), (c) $CH_2O$, (d) PAN, (e) $\Sigma ROONO_2$, and (f) $pNO_3$ over Seoul, South Korea, during
KORUS-AQ. For panels (b) – (f), the observations are after 12:00 local time (03:00 UTC), the
black dots are all data, and the slopes (red line) is an ODR fit to the data. (g – i) Same as Figure 6
for $\Delta OOA/\Delta CO$ versus photochemical age, and including the dilution-corrected production of $O_x$,
$CH_2O$, PAN, $\Sigma ROONO_2$, and $pNO_3$. (g) is over the West Sea (RF12), (h) is over the West Sea
(normal conditions) and (i) is over Seoul. Similar to Figure 4, the $\Delta$ corresponds to subtracting the
background values for the respective species.

$O_x$, $CH_2O$, PAN, $\Sigma ROONO_2$, and $pNO_3$, when dilution-corrected with $\Delta CO$, show a similar

trend as OOA (Figure 7i). From the lowest observed photochemical age (~0.1 equivalent day) to
the highest (~0.85 equivalent day), $O_x$, $CH_2O$, PAN, $\Sigma ROONO_2$, and $pNO_3$ increase by factors of
4, 2, 7, 4, and 2, respectively. Over Mexico City, increases of factors of ~2 ($CH_2O$) (Fried et al.,



2011) and ~3 ($\Sigma$ROONO$_2$) (Perring et al., 2010) were observed, which are comparable to the Seoul
observations. These rapid increases can only be explained by photooxidation of South Korean
primary emissions (hydrocarbons and NO$_x$).

The influence of the upwind, background air masses over the West Sea are investigated

and shown in Figure 7g and h. Over the West Sea, the dilution-corrected concentration of SOA,
O$_x$, CH$_2$O, PAN, $\Sigma$ROONO$_2$, and pNO$_3$ were all nearly constant. This indicates that the secondary
short-lived gas-phase species have reached steady state. Also, since dilution-corrected SOA
concentration is flat, this suggests that the SOA precursors have been depleted, and the SOA
production has ended, with SOA concentration reaching the plateau that is typically observed after
~1 equivalent day (Ortega et al., 2016). The low PAN concentration and influence from transport
over the West Sea was also observed by Lee et al. (2012) over Baengyeoung Island, a regional
background monitoring location for Seoul and South Korea, during August 2010 and March –
April 2011. This further indicates low amounts of PAN are transported due to its thermal
decomposition and very short lifetime, and any production, and correlation, of PAN with OOA
would suggest local, photochemical production.

Besides the ubiquitous (but less specific) secondary species from organic compound

oxidation, OOA shows a robust correlation with dihydroxy toluene (DHT) (Figure 8), a known
SOA precursor from toluene photooxidation (Schwantes et al., 2017). DHT is very short lived,
with a photochemical lifetime of less than 1 hour, and it is formed under both low and high NO
conditions (Schwantes et al., 2017). The lower correlation, compared to the ubiquitous secondary
species, is possibly due to DHT forming from one precursor (toluene) instead of the broad range
of precursors that form OOA and O$_x$, PAN, and CH$_2$O. The correlation of OOA with a known
SOA precursor, that is very short-lived, again supports that OOA production is dominated by



photooxidation of locally emitted hydrocarbons, including toluene. The increasing ratio of OOA
to DHT also suggests that SOA production is fastest at low equivalent ages and is starting to
plateau at higher ages.

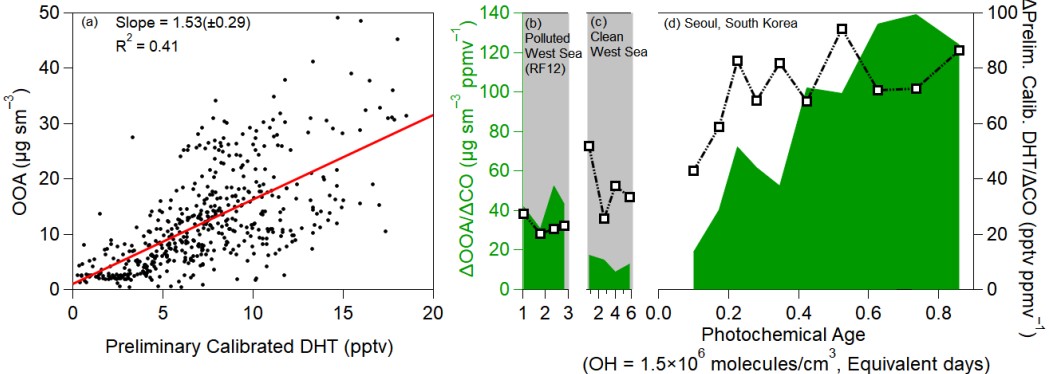


**Figure 8.** (a) Scatter plot of OOA versus DHT over Seoul, South Korea, during KORUS-AQ, after
12:00 local time (03:00 UTC). (b) Same as Figure 7f, but for DHT over the West Sea (RF12). (c)
Same as Figure 7g, but for DHT over the West Sea. (d) Same as Figure 7i, but for DHT over Seoul.
As an important note, concentrations of DHT are based on a preliminary calibration; however, any
further calibrations are not expected to impact the relative trend and general correlation.
The correlation between OOA and secondary species that have very short lifetimes further
suggest that the observed OOA is dominantly due to photooxidation of local emissions to produce
SOA and the other secondary species and not transport. This is due to the fact that the short
photochemical lifetimes of PAN, $CH_2O$, DHT, and $\Sigma ROONO_2$ would cause the secondary species
to be in steady state. The observations over the West Sea, which is mainly upwind of Seoul and
thus background air, show much lower ratios. These two observations further suggest that local
SMA emissions are the precursors that undergo the rapid photooxidation to produce SOA, $pNO_3$,
and the other secondary species.



### 4.2.3 Oxidation state of SOA


We investigate the oxidation state of the observed OA with the van Krevelen diagram

(Heald et al., 2010) in Figure 9a. The slope over Seoul (−0.69) is close to the average slope for
numerous studies summarized by Chen et al. (2015) (−0.60) and similar to the range of slopes
(−0.7 to −1.0) for studies impacted by urban pollution (Aiken et al., 2009; Docherty et al., 2011;
Ge et al., 2012), including Los Angeles during CalNex (ranges from −0.64 to −0.68 from Hayes
et al. (2013) and Ortega et al. (2016)) or chamber studies investigating the photooxidation of
combustion exhausts (Heald et al., 2010; Lambe et al., 2012; Jathar et al., 2013; Presto et al., 2014;
Tkacic et al., 2014; Liu et al., 2015). This generally indicates that the photochemistry controlling
the production of SOA is similar in urban areas, including photooxidation of diesel and gasoline
emissions, evaporative diesel and gasoline, and cooking emissions (Hayes et al., 2015; Woody et
al., 2016; Janssen et al., 2017; Ma et al., 2017; Kim et al., 2018).

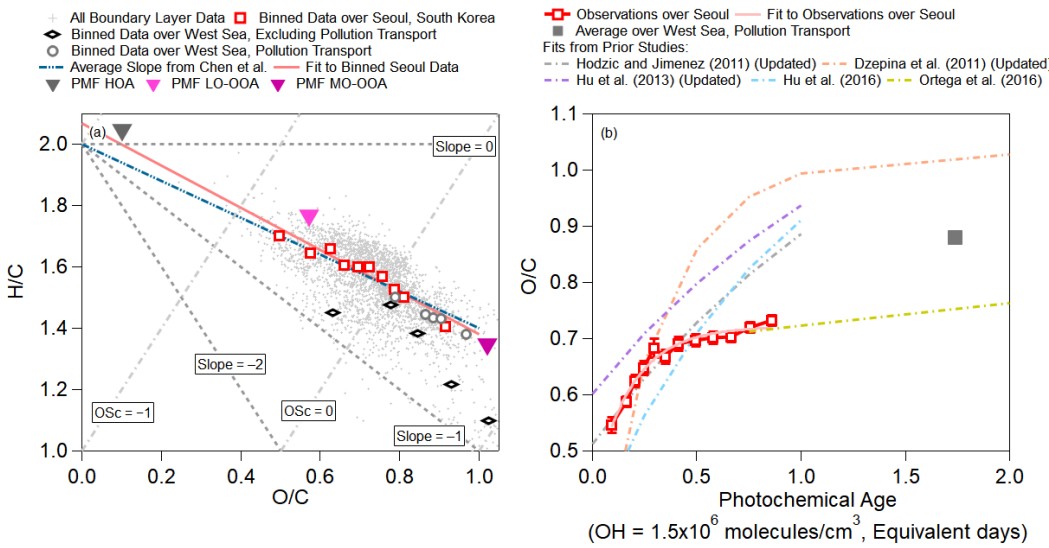


**Figure 9.** (a) Van Krevelen diagram for all of KORUS-AQ. OSc = (O/C − 2×H/C) (Kroll et al.,
2011). The observations are binned, into deciles, for observations over Seoul, South Korea, and
binned, into 5 bins, for clean West Sea, and polluted West Sea. The teal line represents the average
slope reported in Chen et al. (2015) of −0.60, and the light red line represents the slope (slope =



−0.69(±0.15), y-intercept=2.07(±0.11)) observed over Seoul, South Korea, during the campaign.
(b) Binned O/C from observations versus photochemical age, over Seoul, South Korea, and
averaged O/C versus photochemical clock over polluted West Sea. The light red line is the fit to
the observations over Seoul during KORUS-AQ. The values of O/C versus photochemical age
from Hodzic and Jimenez (2011), Dzepina et al. (2011), Hu et al. (2013), are updated with
calibrations of Canagaratna et al. (2015); whereas, Hu et al. (2016), and Ortega et al. (2016) did
not need updates.

The "transport/polluted" evolution of H/C versus O/C falls on the same slope as the

observations over Seoul; however, the values lie at higher O/C ratios, indicative of more aged
OOA. O/C versus H/C during the "transported/polluted" event over the West Sea is also
comparable to the H/C versus O/C slope (−0.63) observed in Chinese outflow at Changdao (Hu et
al., 2013). On the other hand, O/C versus H/C over the West Sea during "clean" events show
distinctly lower values and a steeper evolution (slope = −1.1).

The evolution of O/C with photochemical age over Seoul and over the West Sea is shown

in Figure 9b, along with results from prior studies (Dzepina et al., 2011; Hodzic and Jimenez,
2011; Hu et al., 2013, 2016; Ortega et al., 2016). Note that older studies have been updated with
the calibration of Canagaratna et al. (2015). For the first 0.5 equivalent days, O/C is nearly identical
to the Mexico City observations (Hodzic and Jimenez, 2011); however, after 0.5 equivalent days,
the O/C ratio growth slows down. The O/C evolution then becomes more similar to that observed
when processing Los Angeles air in an OFR (Ortega et al., 2016). The evolution of O/C over Seoul
is at the low end of the range of values observed from prior megacities. The average O/C value
observed over the West Sea during RF12 is more similar to the values observed after 1 equivalent
day in two sites in China (Hu et al., 2013, 2016).
**4.4 Influence of local versus transported SOA precursors to SOA production over Seoul**

OFR results can be used to investigate the role of SOA production from South Korean and

Seoul emissions versus long-distance transported SOA precursors. As shown in prior studies
(Ortega et al., 2016; Palm et al., 2016, 2017, 2018), the SOA potential decreases drastically in the




daytime, as the most reactive compounds to OH have already oxidized and formed SOA. Thus,
these results will not directly capture the full emitted SOA potential for Seoul, South Korea. Also,
recent studies indicate that lower volatility species (e.g., S/IVOCs) can be lost to tubing walls, or
their transfer can be greatly delayed (Pagonis et al., 2017; Deming et al., 2018). Thus, it is likely
that the OFR inlet line on the DC-8 acted at least as a partial sink of S/IVOCs and thus reduced
the measured potential SOA.  As a reminder, a correction is included for the condensational sink
(CS) of LVOC in the OFR based on Eq. (1).

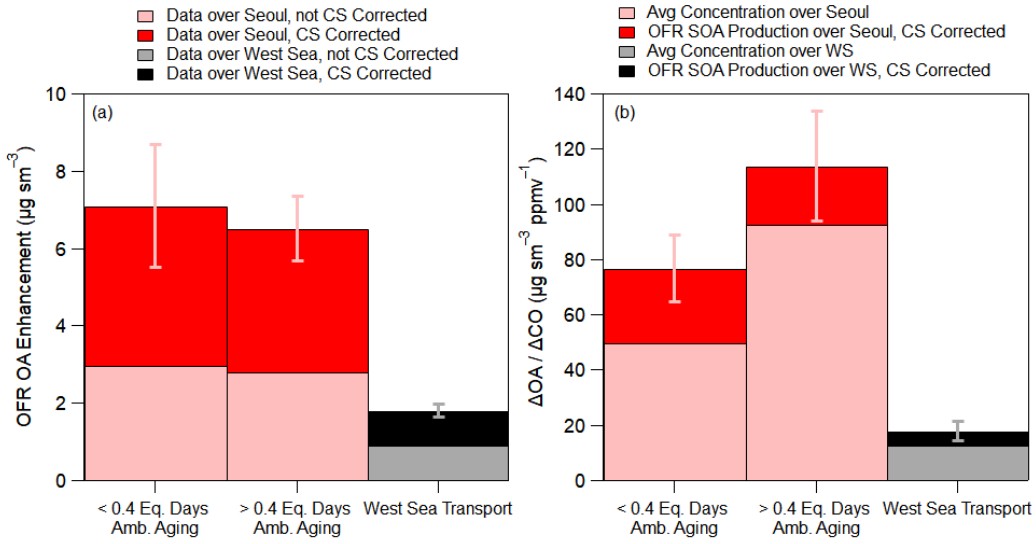


**Figure 10.** (a) Comparison of OFR OA enhancement (OFR OA enhancement = OA exiting OFR
– ambient OA) over Seoul, South Korea, and the West Sea, corrected for evaporation losses. The
difference between the two stacked shaded bars is that the lighter (bottom) shade has no CS
correction whereas the darker shade does. (b) The lighter colored represents the average dilution-
corrected observed ambient OA preexisting concentration corresponding to the OFR observations
of the same air mass. The darker color represents the dilution-corrected SOA production in the
OFR. The average additional photochemical age added in the OFR is ~4 days ($OH_{exp}$ ~ 5.4×10$^{11}$
molecules/cm$^3$×s) for both over Seoul and West Sea. Also, the observations for the West Sea are
for all flights, not including RF12, where the average $\Delta OA/\Delta CO$ was 13 µg sm$^{-3}$ ppmv$^{-1}$. For both
(a) and (b), the OFR observations over Seoul are split between lower and higher equivalent ages
(see x-axis). The average ambient ages for the two bars are 0.17 and 0.63 equivalent days. Error
bars are the standard errors of the observations.



The average OA enhancement in the OFR (OA Enhancement = OA in OFR – Ambient
OA) in Seoul is slightly greater for the less aged ambient air (7.1±1.6 versus 6.5±0.8 µg sm$^{-3}$) but
both values lie within the range of the measurements (Figure 10a). The less aged ambient air show
slightly higher OA enhancement suggests that more SOA precursors might have been present and
available to form SOA mass (Ortega et al., 2016; Palm et al., 2016, 2017, 2018). The OA
enhancement observed over Seoul was a factor of 3.5 greater than observed over the West Sea (~7
µg sm$^{-3}$ over Seoul versus ~2 µg sm$^{-3}$ over the West Sea). The much higher SOA formation
potential observed over Seoul versus the West Sea indicates that the majority of the precursors that
led to the observed SOA and SOA production over Seoul originated from local emissions,
consistent with results above.
Plotting the OA enhancements as ΔOA/ΔCO, similar to Figure 4, the amount of ambient
SOA production, not including pre-existing OA, for ambient air that has aged less than 0.4
equivalent days is 27(±12) µg sm$^{-3}$ ppmv$^{-1}$, a 50% increase compared to the average ΔOA/ΔCO
observed over Seoul at the same ambient photochemical age (Figure 10b). For air older than 0.4
equivalent days, the increase is slightly smaller (21(±20) µg sm$^{-3}$ ppmv$^{-1}$ above ambient pre-
existing OA) since a large fraction of the most reactive, high aerosol producing compounds have
already been depleted and produced ambient SOA (Ortega et al., 2016; Palm et al., 2016, 2017,

2018).

Finally, there is still a small amount of SOA production potential in the air transported over
the West Sea to Seoul. The average potential, not including pre-existing OA, is 5(±4) µg sm$^{-3}$
ppmv$^{-1}$. This is a factor of 4 – 5 less than the potential SOA production observed in the OFR for
Seoul. Including the pre-existing dilution-corrected OA for the West Sea observation (18 µg sm$^{-3}$
ppmv$^{-1}$), the concentration is approximately a factor of 8.5 less than the maximum ambient



dilution-corrected OA concentration and a factor of 6 – 8 less than the total dilution corrected OA
concentration exiting the OFR. Some of this remaining production would have been further
consumed during transport between the West Sea and Seoul (typically 1 day); therefore, it is not
expected to significantly impact the SOA production over Seoul. This further indicates that, during
this campaign, the transported SOA precursors to Seoul from foreign sources did not contribute
significantly to the overall observed SOA production.
**4.5 Calculated precursor contributions to the SOA production over Seoul**

We use a simple SOA model (Dzepina et al., 2009; Zhao et al., 2014) to calculate the

contribution of various precursors to SOA over Seoul (Figure 11, details in Sect. SI 6). Observed
hydrocarbons (Table 2), from WAS, along with estimated S/IVOC (Robinson et al., 2007; Dzepina
et al., 2009; Hayes et al., 2015) and SOA yields updated to account for vapor wall losses (Ma et
al., 2017) were used to estimate (SI Eq. (S3) and (S4)) the contribution of various precursors to
SOA production observed over Seoul. Dzepina et al. (2009) and Hayes et al. (2015) both found
that the "Robinson" parameterization of SOA from S/IVOC was consistent with SOA production
in Mexico City and Los Angeles for observations at 1 equivalent day or less; thus, the same
parameterizations are used here.

The percent difference between the modeled and measured total OA ranged between –24

to 32% with an average value of the observations being 15% higher. This provides confidence that
the calculation described in SI 6 captures the chemical production of SOA over Seoul. Also, the
difference between the estimated and measured OA is comparable to, or better than, found in other
studies that utilized a similar modeling approaches (Dzepina et al., 2009; Zhao et al., 2014, 2016;
Hayes et al., 2015; Huang et al., 2015; Ma et al., 2017), and within the uncertainty of the measured
OA (38%, 2σ).



This box model does not explicitly consider volatile consumer products (VCPs) (Khare
and Gentner, 2018; McDonald et al., 2018), S/IVOC from cooking emissions (e.g., Hayes et al.,
2013; Ots et al., 2016), or glyoxal (Volkamer et al., 2006; Knote et al., 2014), although
contributions from these components may be partially included in the empirical estimation of
S/IVOC. Modeled SOA in Los Angeles, using estimates of S/IVOC from ΔHOA/ΔCO, including
~2/3 of VCPs and not including glyoxal, were able to capture the observed SOA in the first 0.5
equivalent days (Hayes et al., 2015; McDonald et al., 2018), similar to the results here.

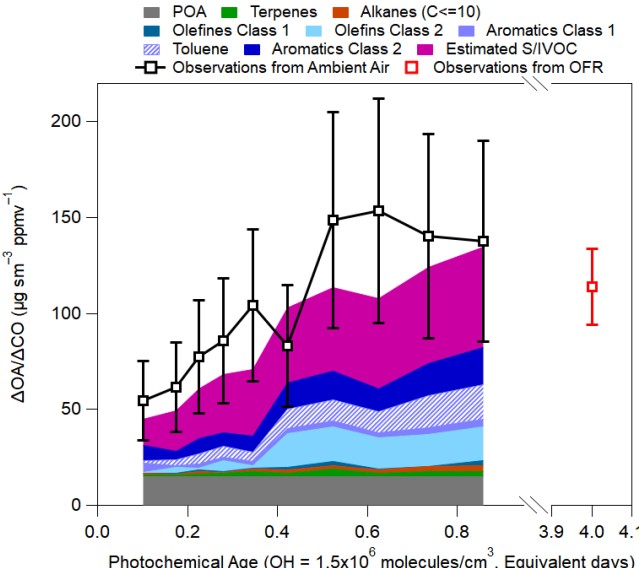


**Figure 11.** Calculated SOA production for KORUS-AQ. POA is from observations shown in
Figure 6, and the observations of ΔOA/ΔCO are from Figure 4. The SOA precursor classes are
defined in Table 2. Note, Toluene is part of Aromatics Class 1 (light purple), but it is shown
separately for discussion. The error bars represent the uncertainty in OA (±38%). The OFR
observations, and error bars, are from Figure 10.



**Table 2.** Definition of classes used in Figure 11. The VOCs listed in the table were all measured by WAS.

| Class | Included Compounds or Parameterization |
|---|---|
| Terpenes | alpha-pinene, beta-pinene |
| Alkanes (C≤10) | methyl-cyclopentane, cyclohexane, methyl-cyclohexane, n-hetpane, n-octane, n-nonane, n-decane, |
| Olefins Class 1 | 1-butene, i-butene, cis-butene, trans-butene |
| Olefins Class 2 | Styrene, 1,3-butadiene |
| Aromatics Class 1 | benzene, toluene, isopropylbenzene, n-propylbenzene, ethylbenzene, |
| Aromatics Class 2 | m+p-xylene, o-xylene, 3-ethyltoluene, 4-ethyltoluene, 1,2,3-trimethylbenzene, 1,2,4-trimethylbenzene, 1,3,5-trimethylbenzene |
| Estimated S/IVOC | $6.7^a \times \Delta HOA/\Delta CO$ ($\Delta HOA/\Delta CO = 22$ µg sm$^{-3}$ ppmv$^{-1}$ from Figure 6) |

[a] This value is taken from Dzepina et al. (2009), which is based on partitioning calculations.

The most important calculated SOA precursors are S/IVOC and the most reactive aromatics (Table 2). These two classes of compounds comprise ~70% of the total modeled SOA over Seoul. The calculation further supports the conclusions from multiple previous studies (Dzepina et al., 2009, 2011; Hodzic et al., 2010; Chen et al., 2015; Hayes et al., 2015; Ma et al., 2017) that aromatics and primary S/IVOC dominate SOA formation over different urban environments. A consistent feature across most species in both classes of compounds is that they all have photochemical lifetimes of less than 1 equivalent day and less than 4 actual hours for the average observed daytime OH ($6 \times 10^6$ molecules/cm$^3$) over Seoul. With the typical wind speeds during KORUS-AQ (~4 m/s), the lifetime of these species would limit their transport approximately 60 km during daytime. Since these compounds have short photochemical lifetimes, and they compose the majority of the calculated SOA budget, our conclusion that the SOA production over Seoul originates from local emissions is further supported.

Numerous prior studies have shown the importance of S/IVOC in order to explain the observed SOA production (Robinson et al., 2007; Dzepina et al., 2009, 2011; Grieshop et al., 2009; Pye and Seinfeld, 2010; Hodzic et al., 2010; Zhao et al., 2014; Jathar et al., 2014; Chen et al., 2015; Hayes et al., 2015; Palm et al., 2016, 2017, 2018; Ortega et al., 2016; Janssen et al., 2017;




Ma et al., 2017). Until recently, it has been analytically challenging to measure these compounds
(Ait-Helal et al., 2014; Zhao et al., 2014; Hunter et al., 2017), and they can make up a small fraction
of the total measured, and speciated, hydrocarbons in an urban location (Ait-Helal et al., 2014;
Zhao et al., 2014). However, due to the higher initial molecular weight, S/IVOC already have a
low saturation concentration ($C^* \sim 1 - 1000$ µg m$^{-3}$ for SVOC and $\sim 1 \times 10^4 - 1 \times 10^6$ for IVOC),
especially compared to aromatic compounds ($C^* \sim 10^7$ µg m$^{-3}$); thus, any addition of functional
groups will more easily lead to the partitioning of oxidized S/IVOC to the particle phase (Robinson
et al., 2007; Hayes et al., 2015; Ma et al., 2017). In urban environments, S/IVOC emissions come
from numerous sources, including transportation, cooking, and VCPs (Robinson et al., 2007;
Hayes et al., 2015; Woody et al., 2016; Janssen et al., 2017; Ma et al., 2017; McDonald et al.,

2018).

The next most important compound is toluene, composing 9($\pm$3)% of the estimated SOA

production. Though this single compound is as important as the rest of Aromatics Class 1, Olefins
Class 1 and 2, alkanes, and terpenes (Table 2) combined, it does not contribute the majority of the
calculated SOA budget, as was recently suggested in another study (Wu et al., 2016). The average
aerosol yield for toluene used in this study ($Y \approx 0.30$) is similar to the value used in Wu et al.
(2016) and recommended by Hildebrandt et al. (2015). The aerosol yield is similar for all
aromatics; however, the more reactive aromatics will contribute more SOA per unit precursor at
shorter photochemical ages. The longer photochemical lifetime (factor of 2) for toluene decreases
the overall amount of SOA produced compared to the very reactive aromatics.
**4.6 Conceptual model representing rapid photochemical production**

A conceptual model representing rapid photochemical production of SOA, pNO$_3$, O$_x$, and

CH$_2$O is presented here. For the model, the flow is simplified to be from the west to the east. The



lateral and vertical dilution have been represented as the equivalent first order rate, constrained by
observations ($\sim$0.7 day$^{-1}$). Also, the hemispheric and foreign transport is accurate on average based
on observations, and is discussed in Sect. 3.3. For the production over Seoul, it is represented by
photochemical aging, constrained by observations (Figure 7i), as a first order rate. Thus, the
important processes are represented with realistic quantitative constraints, but in a simple enough
system to demonstrate the impact of the secondary chemistry from Seoul.

The results are shown in Figure 12. The figure summarizes and demonstrates the results

discussed throughout the paper. First, as discussed in Sect. 3.2 and 4.2, there is no clear net
production of the pollutants over the West Sea. Instead, they undergo dilution as the air travels
across the West Sea. Then, as the air enters SMA area, there are fresh injections of primary
emissions (CO, HOA, hydrocarbons, and NO$_x$). These primary emissions undergo rapid
photooxidation to produce SOA, pNO$_3$, O$_x$, and CH$_2$O, as detailed in Sect. 4.2. As demonstrated
in Figure 12, most of the production occurs with SMA prior to dilution taking over. This
demonstrates that the emissions and subsequent chemistry from SMA are directly impacting the
residents of SMA. Thus, control of the primary pollutants, including the SOA precursors discussed
in Sect. 4.5 (aromatics and S/IVOC), and NO$_x$, would substantially reduce concentration of the
secondary photochemical products impacting SMA, even during period of higher foreign transport
than observed during KORUS-AQ.





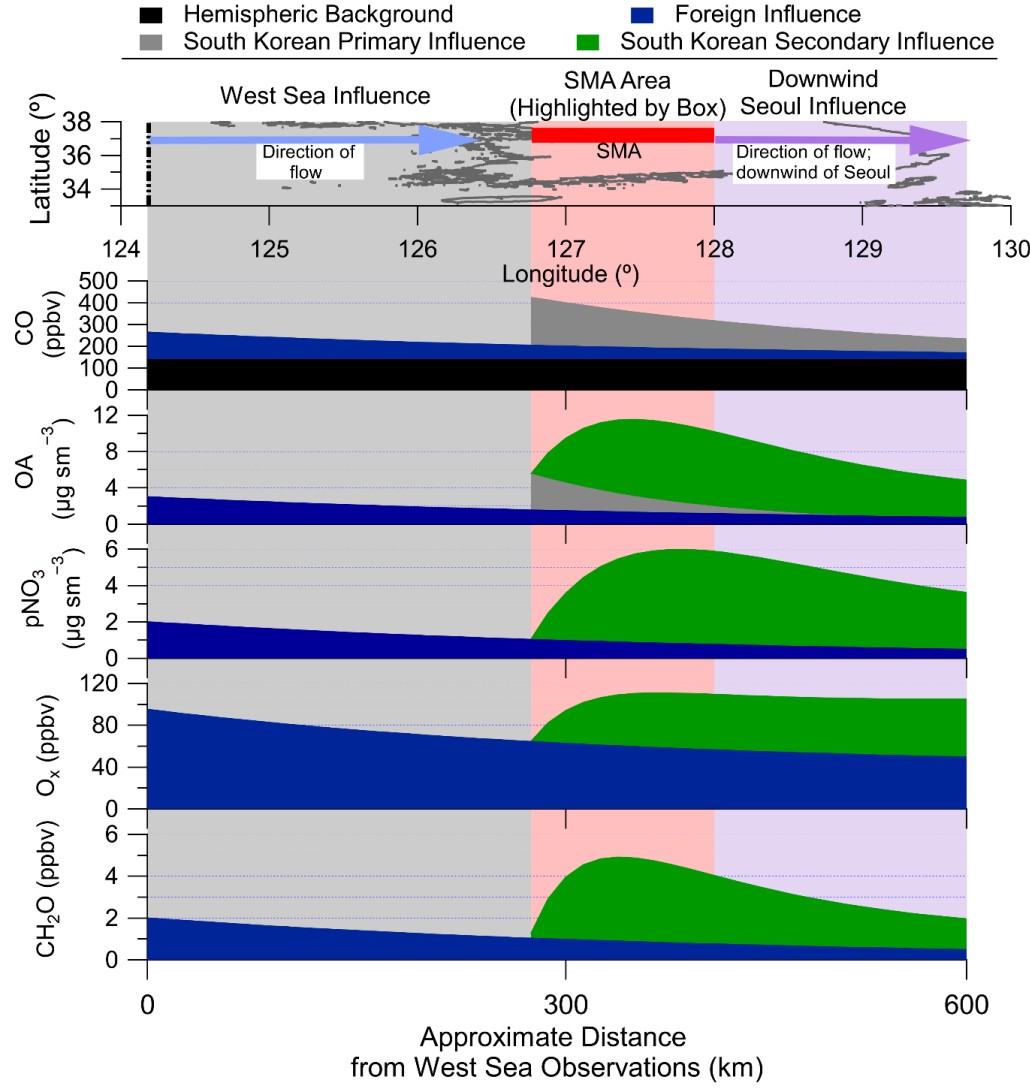


**Figure 12.** Conceptual model representing the transport of background into Seoul, and the emissions of primary species (CO and HOA) and photochemical production of secondary species (SOA, pNO$_3$, O$_x$, and CH$_2$O) impacting Seoul.



**5. Summary**
A suite of aerosol- and gas-phase measurements were made over Seoul and the West Sea
during May and June, 2016, as part of the KORUS-AQ campaign. The results from this study are
summarized below.
(1) Using a combination of a Lagrangian backtrajectory model (FLEXPART) and
observations, the hemispheric CO background was estimated to be 140 ppbv, the CO foreign
background over Seoul was estimated to be 60 ppbv, and the remainder of the CO over Seoul was
due to South Korean emissions. The CO background analysis allows estimating background values
for other species used throughout this study. In particular, the OA background was estimated to be
between $1 - 3 \, \mu g \, sm^{-3}$.
(2) FLEXPART was also used for source analysis of $NO_2$, as a surrogate for SOA
precursors. $NO_2$ has a photochemical lifetime of less than 1 day (similar to the dominant urban
SOA precursors). Results from FLEXPART indicate that greater than 90% of $NO_2$ originates from
South Korea, consistent with most of the important SOA and $pNO_3$ precursors also originating
there.
(3) Factor analysis of OA showed that the OA growth over Seoul was dominated by OOA
(surrogate for SOA). This OOA (background subtracted) was low at low photochemical ages and
rapidly increased throughout the day as photochemistry occurred. This points to local emissions
controlling SOA production over Seoul.
(4) OOA was correlated with secondary gas-phase species, including $O_x$ ($O_3 + NO_2$),
formaldehyde, peroxy acetyl nitrates, sum of acyl peroxy nitrates, dihydroxy toluene, and $pNO_3$.
Correlation with these species indicates that the SOA was produced from local emissions and



photochemistry since some of these compounds (CH$_2$O and PAN) had photochemical lifetimes of
less than three daytime hours during KORUS-AQ.

(5) Using an airborne OFR for the first time, the amount of potential SOA produced from

air sampled over Seoul was a factor of three higher than for air sampled over the West Sea (a
background inflow location). This points to local SOA precursor emissions from Seoul, and
subsequent rapid photochemistry, causing the increase in SOA observed over Seoul. The air
sampled over West Sea did not have enough SOA precursors to cause the SOA production
observed over Seoul.

(6) A simple box model showed good agreement with the measured SOA growth. This

allows an estimation of the contribution of various precursors to SOA over Seoul. Hydrocarbons
with a photochemical lifetime of less than one day dominate the production of SOA. Specifically,
short lived aromatic compounds (i.e., ethyltoluenes, xylenes, trimethylbenzenes) and S/IVOC are
the main precursors to SOA, accounting for 70% of the calculated SOA. Toluene was found to
contribute 9% of the calculated SOA.

(7) Over Seoul, a large megacity with numerous sources of emissions, local emissions and

their photochemical products overwhelm the foreign influence during KORUS-AQ. However, for
smaller cities or more rural areas in South Korea that are not downwind of Seoul or other large
sources, the foreign influence can more easily overwhelm the smaller local emissions. Thus,
outside of the Seoul Metropolitan Area, the foreign influence has a greater impact on the air
quality. During periods in which the foreign influence is larger than during KORUS-AQ (due to
more favorable transport conditions), it will be more comparable to the importance of the Seoul
emissions. However, given the apparently stronger emissions of SOA precursors than in other
megacities, reducing South Korean emissions should improve air quality under all conditions.



**Author Contribution**

BAN, PCJ, DAD, JCS, and JLJ collected the AMS data; BA, AJB, CAC, and KLT collected the data from LARGE; DRB collected the WAS data; WHB collected the OH, $HO_2$, and OHR data; YC and JPD collected the CO measurements from Picarro; JD and ES collected MC/IC and filter measurements; GD and SEP collected $H_2O$ and ambient CO measurements; AF collected $CH_2O$ measurements; LGH collected PAN, PPN, and $SO_2$ measurements; MJK collected $HNO_3$, DHT, and HCN measurements; CK ran the FLEXPART analysis; KDL collected the BC measurements; TL and TP collected the KAMS data; and, JHW provided the emissions for the FLEXPART analysis. JAdG assisted in the analysis of the photochemical clocks and SOA production. BAN and JLJ prepared the original manuscript, and all authors contributed to the review and editing of the manuscript.

**Acknowledgements**

This study was supported by NASA grants NNX15AT96G & 80NSSC18K0630, and EPA STAR 83587701-0. It has not been formally reviewed by EPA. The views expressed in this document are solely those of the authors and do not necessarily reflect those of EPA. EPA does not endorse any products or commercial services mentioned in this publication. Joost de Gouw worked as a part-time consultant for Aerodyne Research Inc during the preparation phase of this manuscript. The authors acknowledge Joshua Schwarz and Anne Perring for the use of their black carbon data, Ronald Cohen, Paul Wooldridge, and Paul Romer for use of their $\Sigma RONO_2$ and $\Sigma ROONO_2$ data, Paul Wennberg and John Crounse for the use of their HCN, DHT, and $HNO_3$ data, Armin Wisthaler for the use of their PTR-MS data, and Andrew Weinheimer and Denise Montzka for the use of their $O_3$, $NO_x$, and $NO_y$ data. Finally, the authors acknowledge the NOAA ESRL GMD CCGG for the use of the CO measurements in Mongolia and China for the background measurements.

**Data Availability**

Measurements and FLEXPART results from the KORUS-AQ campaign are available at https://www-air.larc.nasa.gov/cgi-bin/ArcView/korusaq. Measurements for the CO background are available at ftp://aftp.cmdl.noaa.gov/data/trace_gases/co/flask/surface/.

**Competing Interests**

The authors declare that they have no conflict of interest.



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
