# Peer review of "Secondary Organic Aerosol Production from Local Emissions Dominates the"

_Atmospheric Chemistry and Physics, 2018_

## Referee Comment (RC1) · Anonymous Referee #1 · 14 Sep 2018

Secondary Organic Aerosol Production from Local Emissions Dominates the Organic Aerosol Budget over Seoul, South Korea, during KORUS-AQ

This paper presents airborne observations of organic aerosol (OA) made over and near Seoul, Korea and investigates the local urban contribution to the formation of secondary organic aerosol (SOA). The upwind transport of OA and SOA precursors coming into the Seoul region from China are evaluated to isolate the impact of transport versus local emissions and SOA production. The formation of SOA was determined to be mostly locally produced through evaluation using FLEXPART source analysis and factor analysis of OA and their correlation with other short-lived photochemically produced species. In addition, results using an OFR indicated greater potential SOA formation in air sampled over Seoul compared with air sampled upwind. Box model calculations reproduced SOA within 15% and showed that short-lived aromatic hydrocarbons are the main SOA precursors.

This is a relevant paper for ACP and would be of interest to ACP readers. The paper is very comprehensive, well written with clear study objectives, logically presented and articulated conclusions.

I recommend acceptance to ACP after addressing some minor comments:

L184: not sure what is meant by 'also' encountered? In addition to what? Same thing with 'also in the next line.

L192: highly customized, high-resolution time-of-flight mass spectrometer?? What is customized as it seems pretty standard.

L211: Are there any particle losses through the pressure controlled inlet, and if so, how were they accounted for?

L230: ensure quality control…..this oversimplifies the need for particle filter measurements. Suggest a brief explanation.

L240: should state range of CE determined

L245: Was the contribution from organic nitrates estimated in this study (and stated anywhere)?

L249: What was scaled? The detection limits? Why were they scaled?

L252-257: Should provide brief context behind these statements ie. 24 hour power was not available and the AMS needed to be restarted each day (flight) etc. Otherwise confusing.

L257: Why reference another paper for the accuracy of your measurements? What are the estimated uncertainties related to subtracting an appropriate background (chopper closed) when sampling in/out of plume. How wide are the regional plumes? More than 1 minute? How does this affect the accuracies?

L323-325: I presume the pNO3 and OA measurements are also affected for those data taken not from the OFR?

L611-612: Awkward wording

L621-622: maximum Chinese outflow stated at 40 ug/m3/ppmv, but this is only based on one flight….? I would think there could be substantial variation in this.

L611-616: The results do seem to depend on assumed CO background. This is apparent in Fig S34 where ΔOA/ΔCO is different with different CO backgrounds especially in more aged air masses. What I think the authors mean is that there is still significant and rapid SOA formation greater than many megacities compared in the Figure, and therefore doesn't change the interpretation of this comparison. Also in SI Fig 35, confused about what is being shown here – I presume this is for the Seoul data, but it's also not clear what CO background is being used here because the text in the paragraph indicates a CO background of 140ppbv, but the Figure 4 captions indicates for the Seoul data a background CO=200ppbv was applied (140ppb was for the WS).

L606-607: Seems to me that SOA formation is actually much higher than LA and Mexico City, rather than similar to.

L623: Do you mean finding SOA greater than POA concentrations….? And confused as to where this is shown in Figure 4 with the current data set?

Fig 7g,h: Figh is discussed, but it is interesting why in the more polluted air masses over the West Sea, the CH2O and PAN ratios to CO are lower and constant compared to the West Sea clean.

Supplemental, L57: What is meant by 'built-in' DMA?

Supplemental, L65: Do you mean mobility diameter as identified on L57?

Supplemental, L72: Was the organic NO3 contribution determined? Didn't see this in the main text

Supplemental, Fig 2: Figure indicates eptof 3-5 sec and text indicates 8s…? I'd like to see the figure caption explain the ambient closed data in that there is a line connecting the 6s of data and the ambient open-closed are corrected through an interpolated background.

Supplemental, SI Fig 37: Light pink and dark pink squares look almost identical; please change colour of one. Not sure where the light red crosses are shown for the binned values.

Typos:

L102: should be 'suggest to be major' ie. Remove 'a'

L155: should be instruments not instrument

L177: the wind 'was' instead of 'is'

L251: periodic not period

Fig 10: olefins, not olefines

L933: units needed

L207: ram should be RAM as it is an acronym

Fig 10(b): lighter color, not lighter colored

---

## Referee Comment (RC2) · Anonymous Referee #2 · 6 Nov 2018

Nault et al. present the findings of organic aerosol measurements collected during the KORUS-AQ field campaign. The work finds that the secondary organic aerosols (SOA) formed in Seoul are predominantly formed from SOA precursors emitted locally. This conclusion is supported by back trajectory modeling, measurements of other secondary species (e.g. formaldehyde), airborne oxidation flow reactor measurements, and box model simulations. The paper is generally well written and the work provides valuable insights into SOA. I recommend the paper be accepted with a few minor revisions detailed below.

[Figure]

General Question: Perhaps this is beyond the scope of this work, by do you have insights into why more SOA is formed from Seoul compared to other cities presented in Fig. 4? Is it just more reactive since other rapidly oxidized species follow a similar trend (Fig. 7) or are there more SOA precursors in Seoul or perhaps something else entirely which is unique to Seoul?

Line 137: Add "is" after "aerosol load"

Line 256: Remove "much" and change "for several hours" to "after several hours"

Line 264: Change "allows the measurement of" to "measures"

Line 511: It's unclear how the dilution rate is used to calculate the 60 ppbv of $CO_{foreign}$. Perhaps this is covered in one of the other studies referenced?

Lines 520: Hemispheric background not being included in the figure is a bit misleading since it is included in 2b and 2c. Presumably, the exclusion applies only to 2a and 2d but this is not apparent in Line 520.

Line 526: Is "background subtracted CO" not the same as "$CO_{SouthKorea}$"?

Line 529: I may be misinterpreting this, but because $CO_{foreign}$ is both in the numerator and the denominator, $OA_{background}$ would simply equal OA, which doesn't seem right.

Line 530: How were the fractions of HOA, LO-OOA, and MO-OOA determined?

Line 539: I had a hard time following this part of the sentence. I believe what's plotted is the FLEXPART $NO_2$ but the "sampled from aircraft position for contributions to" part is unclear.

---

## Author Comment (AC1) · 17 Nov 2018

**Response to reviewers' comments on the paper "Secondary Organic Aerosol Production from Local Emissions Dominates the Organic Aerosol Budget over Seoul, South Korea, during KORUS-AQ"**

We would like to thank both reviewers for their time and for their useful comments, that have helped us improve and clarify our paper. For ease, comments from reviewers are in black, responses in blue, and new text added to paper in **bold blue**.

*Reviewer #1*

This paper presents airborne observations of organic aerosol (OA) made over and near Seoul, Korea and investigates the local urban contributions to the formation of secondary organic aerosol (SOA). The upwind transport of OA and SOA precursors coming into the Seoul region from China are evaluated to isolate the impact of transport versus local emissions and SOA production. The formation of SOA was determined to be mostly locally produced through evaluation using FLEXPART source analysis and factor analysis of OA and their correlation with other short-lived photochemically produced species. In addition, results using an OFR indicated greater potential SOA formation in air sampled over Seoul compared with air sampled upwind. Box model calculations reproduced SOA within 15% and showed that short-lived aromatic hydrocarbons are the main SOA precursors.

This is a relevant paper for ACP and would be of interest to ACP readers. The paper is very comprehensive, well written with clear study objectives, logically presented and articulated conclusions.

I recommend acceptance to ACP after addressing some minor comments.

R1.1: L184: not sure what is meant by 'also" encountered? In addition to what? Same thing with 'also' in the next line.

In line 184, we were referring to the three jet ways encountering pollution similar to Seoul. We have clarified line 184 to say:

**"Many of the 3 lower elevation sampling legs around South Korea encountered significant pollution, similar to the flight segments over Seoul."**

In line 185, we were referring to the coordinates for the three legs, similar to Seoul, being located in Table S2. We have updated line 185 to say:

**"Similar to Seoul, the approximate coordinates defining these regions are included in Table S2."**

R1.2: L192: highly customized, high-resolution time-of-flight mass spectrometer?? What is customized as it seems pretty standard.

How the CU-AMS is highly customized is discussed throughout Section 2.2 and highlighted here: (1) use of a customized pressure control inlet (PCI, line 211 - 212); (2) operation of CU-AMS in Fast Mass Spectrum (line 222 - 225); rapid particle filtering (line 228 - 230); and, use of a cryogenic pump to improve limits-of-detection for many species (line 252 - 257). There are also additional updates that are not described in detail, such as automatic inlet flow control vs. altitude, improvements to the supplementary data acquisition hardware and software, etc. These items are not part of a standard Aerodyne HR-ToF-AMS and have required an extensive amount of work and testing and improvement over many campaigns. These updates allow this instrument to measure NR-PM$_1$ on aircraft with fast time response (1 s), low detection limits, and improved accuracy, which is necessary to measure NR-PM$_1$ with high sensitivity at high altitude (including the customized PCI) and remote regions.

R1.3: L211: Are there any particle losses through the pressure controlled inlet, and if so, how were they accounted for?

The particle transmission (both ammonium nitrate and ammonium sulfate) and ionization efficiency (IE) were calibrated through the entire plumbing and pressure-control inlet. As was already shown in SI Figure 4, there were minimal losses of ammonium nitrate and sulfate through the pressure-controlled inlet compared to an instrument without a pressure-controlled inlet (the Hu et al. (2017) line in the figure).

We added the following statement to L213:
**"The lens transmission calibration (SI Sect. 2 and Fig. S4) was conducted through the entire plumbing, including the pressure-controlled inlet. There were minimal losses of ammonium nitrate and sulfate through the pressure-controlled inlet during these calibrations."**

R1.4: L230: ensure quality control . . . this oversimplifies the need for particle filter measurements. Suggest a brief explanation.

We have added the following explanation to L234 (updated text):

**"The filters provide data quality checks throughout the flight by checking for leaks as the cabin changes pressure, to determine the response time of the different species (typically less than 2 seconds), and to validate the real-time continuous detection limits calculated by the Drewnick et al. (2009) method."**

R1.5: L240: should state range of CE determined

We added the following text (L248 updated text):

**"The collection efficiency (CE) for the CU-AMS was estimated per Middlebrook et al. (2012), and ranged from 0.5 – 1 (Fig. S28), with most of the values occurring around 0.5."**

R1.6: L245: Was the contribution from organic nitrates estimated in this study (and stated anywhere)?

We added the following text to L254 (updated text):

**"On average, during the campaign, organic nitrates were ~8% of the total CU-AMS NO₃ signal, and were only an important contribution to the NO₃ signal when NO₃ was less than 0.50 µg sm⁻³ (Figure S38)."**

And the following supplemental figure has been added to the paper:

[Figure]

**Figure 1. (a) Time series of the fractional contribution of organic nitrates (pRONO₂) to the total pNO₃ signal during KORUS-AQ. (b) Fractional contribution of organic nitrates versus pNO₃ during KORUS-AQ.**

R1.7: L249: What was scaled? The detection limits? Why were they scaled?

The detection limits were scaled because the filters takes into account all ions for the different NR-PM₁ whereas Drewnick et al. (2009) only uses some ions to determine the detection limit.
We changed the sentence (L259 updated text) to say:

**"The detection limits were scaled by ~×0.8, based on comparisons with periodic filter blanks, since the Drewnick et al. (2009) method only uses some ions to determine the detection limits, while filters take all ions into account, and thus the latter provide a more accurate estimate."**

R1.8: L252-257: Should provide brief context behind statement ie. 24 hour power was not available and the AMS needed to be restarted each day (flight) etc. Otherwise confusing.

We added the following text (L269 in new text):

**"The cryogenic pump is necessary since the airplane had power only 3 hours prior to take-off until 2 hours after landing; therefore, the CU-AMS was not constantly being pumped. This leads to high backgrounds each time the instrument is started."**

R1.9: L257: Why reference another paper for the accuracy of your measurements? What are the estimated uncertainties related to subtracting an appropriate background (chopper closed) when sampling in/out of plumes? How wide are the regional plumes? More than 1 minute? How does this affect accuracies?

The reviewer may be confusing precision versus accuracy. Precision can be evaluated with changes in background, especially for sampling in and out of plumes, sampling through a particle filter (R1.4), and calibrations. We already report the limits of detection, which is related to precision, in line 241 - 243.

Accuracy is controlled by the uncertainties in ionization efficiency of nitrate (IE), relative ionization efficiencies of the different species (RIE), and collection efficiency (CE), of which, only a few can be regularly calibrated (IE and RIE for ammonium, sulfate, and chloride). The accuracy due to all three effects needs to be propagated in the quantification equation to determine the full accuracy of the AMS. Bahreini et al. (2009, their Supp. Info) performed this detailed analysis to determine what the overall accuracy (which typically dominates uncertainty, as precision tends to be much better) of the AMS. Due to this, the accuracy Bahreini et al. (2009) determined is regularly used and cited for the AMS.

R1.10: L323-325: I presume the pNO3 and OA measurements are also affected for those data taken not from the OFR?

We have shown through numerous studies that the short residence time to sample the measurements (~0.4 s in boundary layer and ~1.0 s at 7500 m) does not lead to any evaporation of OA and $pNO_3$ (Guo et al., 2016, 2017; Shingler et al., 2016).

We have added the following text in line 340 (new text):
**" . . . and longer residence times (~150 s). However, for ambient measurements, the residence time was less than 1 s (Sect 2.2), which is rapid enough to prevent volatilization of OA and $pNO_3$, as discussed in prior work (Guo et al., 2016, 2017; Shingler et al., 2016)."**

R1.11: L611-612: Awkward wording.

Please see response to R1.13.

R1.12: L621-622: maximum Chinese outflow stated at 40 µg/m3/ppmv, but this is only based on one flight….? I would think there could be substantial variation in this.

As we discussed in Lines 619 - 622, dilution-corrected OA concentration of 40 µg/m3/ppmv was observed in China (Beijing, Hu et al. (2016)) and downwind of China (Hu et al., 2013). This is in agreement with the value observed over the West Sea during the "transport/polluted" event. We have updated the text (L647 - 650 new text) to say:

**"Since 40 µg m$^{-3}$ ppmv$^{-1}$ has been observed in and downwind of China (Hu et al., 2013, 2016), and observed over the West Sea during the "transport/polluted" event, this dilution-**

**corrected OA concentration is taken to be representative of transport events from China to South Korea.**"

R1.13: L611-616: The results do seem to depend on assumed CO background. This is apparent in Fig. S34 where ΔOA/ ΔCO is different with different CO backgrounds especially in more aged air masses. What I think the authors mean is that there is still significant and rapid SOA formation greater than many megacities compared in the Figure, and therefore doesn't change the interpretation of this comparison. Also in SI Fig 35, confused about what is being shown here – I presume this is for Seoul data, but it's also not clear what CO background is being used here because the text in the paragraph indicates a CO background of 140ppgv, but the Figure 4 captions indicates for the Seoul data a background CO=200ppbv was applied (140ppb was for the WS).

We have rephrased lines 636 (updated text) to say:

**"Finally, though the absolute ΔOA/ΔCO value changes depending on background CO used, assuming a lower CO background does not change the general result that Seoul has higher and more rapid SOA production than has been observed in prior megacities."**

We have made the caption for SI Figure 35 more explicit to say:

**"Same as Figure 4. ΔOA/ΔCO, using CO background of 200 ppbv, versus photochemical age, for observations collected over Seoul. The main difference is that the ΔOA/ΔCO versus different photochemical ages (NO$_x$ clock in green, m+p-xylene/ethylbenzene in blue, and o-xylene/ethylbenzene in black) are plotted to show minimal differences in the final results of ΔOA/ΔCO versus photochemical age."**

R1.14: L606-607: Seems to me that SOA formation is actually much higher than LA and Mexico City, rather than similar to.

We have changed L629 – 631 (updated text) to say:

**"Qualitatively, the time scale for the production and plateauing of dilution-corrected OA is similar for Seoul, Los Angeles, and Mexico City; however, the amount of OA produced per CO is larger for Seoul compared to Los Angeles and Mexico City."**

R1.15: L623: Do you mean finding SOA greater than POA concentrations…? And confused as to where this is shown in Figure 4 with the current data set?

We have rephrased this text (L653 updated text) as:

**"The fact that OA concentrations are greater than POA concentrations at the youngest photochemical ages…"**

R1.16: Fig 7g,h: Figh is discussed, but it is interesting why in the more polluted air masses over the West Sea, the CH2O and PAN ratios to CO are lower and constant compared to the West Sea clean.

We agree this an interesting feature; however, we do not know why this feature occurred. As we are using those species as general tracers of photochemically-produced pollution and not investigating their sources and sinks quantitatively, we prefer to not speculate about the causes of this observation.

R1.17: Supplemental, L57: What is meant by 'built-in' DMA?

We have replaced the text "sized with a built-in differential mobility analyzer" with the following text for clarity:

**"sized with a differential mobility analyzer, TSI model 3080, that was installed in the same rack as the AMS"**

R1.18: Supplemental, L65: Do you mean mobility diameter as identified on L57?

Corrected to **"mobility diameter"**.

R1.18 Supplemental, L72: Was the organic $NO_3$ contribution determined? Didn't see this in the main text.

We have included text and a figure on this. See response to comment R1.6 above.

R1.19: Supplemental, Fig 2: Figure indicates eptof 3-5 sec and text indicates 8s…? I'd like to see the figure caption explain the ambient closed data in that there is a line connecting the 6s of data and the ambient open-closed are corrected through an interpolated background.

Corrected the caption to say:

**"Though the final 8 s of each minute are dedicated to ePToF, some of the time is used by the computer in saving the 6 s of closed and 46 s of open signal and ePToF signal.; therefore, only 3 – 5 s of ePToF signal is actually recorded. The approximate saving times are shown as white spaces."**

Figure has been updated and shown below:

[Figure]

R1.20: Supplemental, SI Fig 37: Light pink and dark pink squares look almost identical; please change colour of one. Not sure where the light red crosses are shown for the binned values.

Updated the figure to make it more readable as:

[Figure]

Updated the caption to read:

**". . . . The quantile average values (averaged the x variables according to quantiles of the y variables) for each comparison are shown in light red circles."**

Typos:
R1.21:  L102: should be 'suggest to be major' ie Remove 'a'

Corrected

R1.22: L155: should be instruments not instrument

Corrected

R1.23: L177: the wind 'was' instead of 'is'

Corrected

R1.24: L251: periodic not period

Corrected

R1.25: Fig 10: olefins, not olefins

Corrected

R1.26: L933: units needed

Corrected

R1.27: L207: ram should be RAM as it is an acronym

Corrected

R1.28: Fig 10(b): lighter color, not lighter colored

Corrected

*Reviewer #2*

Nault et al. present the findings of organic aerosol measurements collected during the KORUS-AQ field campaign. The work finds that the secondary organic aerosols (SOA) formed in Seoul are predominantly formed from SOA precursors emitted locally. This conclusion is supported by back trajectory modeling, measurements of other secondary species (e.g., formaldehyde), airborne oxidation flow reactor measurements, and box model simulations. The paper is generally well written and the work provides valuable insights into SOA. I recommend the paper to be accepted with a few minor revisions detailed below.

R2.1: General question: Perhaps this is beyond the scope of this work, by do you have insight into why more SOA is formed from Seoul compared to other cities presented in Fig. 4? Is it just more reactive since other rapidly oxidized species follow a similar trend (Fig. 7) or are there more SOA precursors in Seoul or perhaps something else entirely which is unique to Seoul?

We agree that this is interesting. We are currently investigating the possible causes for higher SOA formation in Seoul compared to other megacities. This is a very complex topic, whose results may be reported in a future paper.

R2.2: Line 137: Add "is" after "aerosol load"

Corrected

R2.3: Line 256: Remove "much" and change "for several hours" to "after several hours"

Corrected

R2.4: Line 264: Change "allows the measurement of" to "measures"

Corrected

R2.5: Line 511: It's unclear how the dilution rate is used to calculate the 60 ppbv of $CO_{foreign}$. Perhaps this is covered in one of the other studies referenced?

We took the equation $C(t)=C(0) \times exp(-k_{dil} \times t)$, where $t$ = ~1 day, $-k_{dil}$ = ~0.7 day$^{-1}$, and $C(0)$ = 125 ppbv, which leads to ~60 ppbv of foreign CO. If we take a more complicated equation listed in McKeen et al. (1996), which takes the background CO (140 ppbv) into account, we get the exact same answer.

We have added the following text (L528 updated text) for clarification:
**"$CO_{foreign}$ over Seoul was determined by Eq. (3), where $t$ = ~1 day, $-k_{dil}$ = ~0.7 day$^{-1}$, and $C(0)$ = 125 ppbv. Using the full equation from McKeen et al. (1996), a similar value of 60 ppbv $CO_{foreign}$ is derived.**

**$C(t)=C(0) \times exp(-k_{dil} \times t)$                                (3)"**

R2.6: Line 520: Hemispheric background not being included in the figure is a bit misleading since it is included in 2b and 2c. Presumably, the exclusion applies only to 2a and 2d but this is not apparent in Line 520.

We have changed the caption to say:

**"Also, note that FLEXPART does not include hemispheric background; therefore, it is not included in (a) and (d); however, it is included for the actual observations in (b) and (c)."**

R2.7: Line 526: Is "background subtracted CO" not the same so "$CO_{South\ Korea}$"?

We have changed "background subtracted CO" to the **"hemispheric background-subtracted CO"**, as this is for observations over the West Sea and not over Seoul, yet.

R2.8: Line 529: I may be misinterpreting this, but because $CO_{foreign}$ is both in the numerator and the denominator, $OA_{background}$ would simply equal OA, which doesn't seem right.

To avoid this point of confusion, we have changed the equation from $OA_{background} = CO_{foreign} \times (OA/CO)_{foreign}$ to:

$$OA_{background}(t) = CO_{foreign}(t) \times (OA/CO)_{foreign}(0)$$

where t is for any observations after the initial OA/CO observation.

R2.9: Line 530: How were the fractions of HOA, LO-OOA, and MO-OOA determined?

HOA, LO-OOA, and MO-OOA were determined by positive matrix factorization, and the background values were determined similarly to OA, as described comment R2.8.

R2.10: Line 539: I had a hard time following this part of the sentence. I believe what's plotted is the FLEXPART $NO_2$ but the "sampled from aircraft position for contributions to" part is unclear.

We have removed the "sampled from aircraft position" for clarity.

*Other Edits*

While working on the comments, we noticed an error in Fig. 11. The HOA was low by 7 µg sm$^{-3}$ ppmv$^{-1}$. We have updated the plot (see below). It did not change the conclusions, but it changed the average percent difference from 15% to 11%.

[Figure]

**References**

Bahreini, R., Ervens, B., Middlebrook, A. M., Warneke, C., de Gouw, J. A., DeCarlo, P. F., Jimenez, J. L., Brock, C. A., Neuman, J. A., Ryerson, T. B., Stark, H., Atlas, E., Brioude, J., Fried, A., Holloway, J. S., Peischl, J., Richter, D., Walega, J., Weibring, P., Wollny, A. G. and Fehsenfeid, F. C.: Organic aerosol formation in urban and industrial plumes near Houston and Dallas, Texas, J. Geophys. Res., 114(16), D00F16, doi:10.1029/2008JD011493, 2009.

Drewnick, F., Hings, S. S., Alfarra, M. R., Prevot, A. S. H. and Borrmann, S.: Aerosol quantification with the Aerodyne Aerosol Mass Spectrometer: detection limits and ionizer background effects, Atmos. Meas. Tech., 2(1), 33–46, doi:10.5194/amt-2-33-2009, 2009.

Guo, H., Sullivan, A. P., Campuzano-Jost, P., Schroder, J. C., Lopez-Hilfiker, F. D., Dibb, J. E., Jimenez, J. L., Thornton, J. A., Brown, S. S., Nenes, A. and Weber, R. J.: Fine particle pH and the partitioning of nitric acid during winter in the northeastern United States, J. Geophys. Res. Atmos., 121(17), 10,355-10,376, doi:10.1002/2016JD025311, 2016.

Guo, H., Liu, J., Froyd, K. D., Roberts, J. M., Veres, P. R., Hayes, P. L., Jimenez, J. L., Nenes, A. and Weber, R. J.: Fine particle pH and gas–particle phase partitioning of inorganic species in Pasadena, California, during the 2010 CalNex campaign, Atmos. Chem. Phys., 17(9), 5703–5719, doi:10.5194/acp-17-5703-2017, 2017.

Hu, W., Hu, M., Hu, W., Jimenez, J. L., Yuan, B., Chen, W., Wang, M., Wu, Y., Chen, C., Wang, Z., Peng, J., Zeng, L. and Shao, M.: Chemical composition, sources and aging process of sub-micron aerosols in Beijing: contrast between summer and winter, J. Geophys. Res. Atmos., doi:10.1002/2015JD024020, 2016.

Hu, W., Campuzano-Jost, P., Day, D. A., Croteau, P., Canagaratna, M. R., Jayne, J. T., Worsnop, D. R. and Jimenez, J. L.: Evaluation of the new capture vapourizer for aerosol mass spectrometers (AMS) through laboratory studies of inorganic species, Atmos. Meas. Tech., 10(6), 2897–2921, doi:10.5194/amt-10-2897-2017, 2017.

Hu, W. W., Hu, M., Yuan, B., Jimenez, J. L., Tang, Q., Peng, J. F., Hu, W., Shao, M., Wang, M., Zeng, L. M., Wu, Y. S., Gong, Z. H., Huang, X. F. and He, L. Y.: Insights on organic aerosol aging and the influence of coal combustion at a regional receptor site of central eastern China, Atmos. Chem. Phys., 13(19), 10095–10112, doi:10.5194/acp-13-10095-2013, 2013.

McKeen, S. A., Liu, S. C., Hsie, E.-Y., Lin, X., Bradshaw, J. D., Smyth, S., Gregory, G. L. and Blake, D. R.: Hydrocarbon ratios during PEM-WEST A: A model perspective, J. Geophys. Res. Atmos., 101(D1), 2087–2109, doi:10.1029/95JD02733, 1996.

Middlebrook, A. M., Bahreini, R., Jimenez, J. L. and Canagaratna, M. R.: Evaluation of Composition-Dependent Collection Efficiencies for the Aerodyne Aerosol Mass Spectrometer using Field Data, Aerosol Sci. Technol., 46(3), 258–271, doi:10.1080/02786826.2011.620041, 2012.

Shingler, T., Crosbie, E., Ortega, A., Shiraiwa, M., Zuend, A., Beyersdorf, A., Ziemba, L., Anderson, B., Thornhill, L., Perring, A. E., Schwarz, J. P., Campazano-Jost, P., Day, D. A., Jimenez, J. L., Hair, J. W., Mikoviny, T., Wisthaler, A. and Sorooshian, A.: Airborne characterization of subsaturated aerosol hygroscopicity and dry refractive index from the surface to 6.5km during the SEAC4RS campaign, J. Geophys. Res., 121(8), 4188–4210,

doi:10.1002/2015JD024498, 2016.